COMMUNICATIONS

# A recent increase in global wave power as a consequence of oceanic warming

Borja G. Reguero[1,2], Iñigo J. Losada[1] & Fernando J. Méndez[1]

Wind-generated ocean waves drive important coastal processes that determine flooding and erosion. Ocean warming has been one factor affecting waves globally. Most studies have focused on studying parameters such as wave heights, but a systematic, global and long-term signal of climate change in global wave behavior remains undetermined. Here we show that the global wave power, which is the transport of the energy transferred from the wind into sea-surface motion, has increased globally (0.4% per year) and by ocean basins since 1948. We also find long-term correlations and statistical dependency with sea surface temperatures, globally and by ocean sub-basins, particularly between the tropical Atlantic temperatures and the wave power in high south latitudes, the most energetic region globally. Results indicate the upper-ocean warming, a consequence of anthropogenic global warming, is changing the global wave climate, making waves stronger. This identifies wave power as a potentially valuable climate change indicator.

[1] Environmental Hydraulics Institute "IH Cantabria", Universidad de Cantabria, 39011 Santander, Cantabria, Spain. [2] Present address: Institute of Marine Sciences, University of California, 115 McAllister Way, Santa Cruz, CA 95060, USA. Correspondence and requests for materials should be addressed to B.G.R. (email: borjagreguero@gmail.com)

Climate change is modifying oceans in different ways, including ocean atmosphere circulation and water warming[1]. The effects of climate change will particularly be present at the coast, where humans and oceans meet[2]. Surface gravity waves generated by winds have far-reaching implications for coastal areas. Wind-waves are an important contributor to coastal flooding and sediment transport, shaping headlands, bays and the open coasts, and determining where and how coastal infrastructure are built. Understanding how waves change is critical to assess the impacts of climate change at the coast[3,4]. Waves result from the interaction between the atmosphere and the ocean and therefore are affected by climate change[5–9]. However, a systematic, global and long-term signal of climate change in global wave behavior, similar to the global time series of sea level rise or temperature warming, remains currently undetermined.

Most analyses of the global wave climate (i.e., the description of wave characteristics over a period of time) have focused on identifying historical trends in mean and extreme values of wave parameters, such as wave heights. Wave heights have been increasing in recent decades, particularly at the high latitudes of both hemispheres. Increases have been larger for the extreme values as compared to the mean conditions[7–10]. Satellite-based altimeter measurements from 1985 to 2008 reveal increases of 0.25% per year for the 90th wave height percentile and 0.50% per year for the 99th percentile, in both hemispheres[9]. Hindcast data also show significant increases in extreme wave heights at the high latitudes of the southern hemisphere[11], 0.25–0.9% per year for the 90th percentile in the north Atlantic and the north Pacific, and decreases in the mid-latitudes[8,12]. However, not only have wave heights been changing, wave periods have also increased[13], and direction of waves shifted, for example in the southern ocean[11] and in the north Atlantic[12]. However, changes in the global wave energy have received less attention, particularly in the context of climate change[14]. Despite the changes detected in different wave parameters, a global and long-term time series of the effect of climate change in the global wave climate remains undetected.

Increase in wave heights (and wave energy) is primarily driven by increases in surface wind energy[15]. Wave heights have been increasing associated to upward trends in global sea surface mean and extreme wind speeds across ocean basins from 1988 to 2011[9,16,17]. However, interannual climate patterns, like the Southern Annular Mode (SAM) or the Pacific Decadal Oscillation, drive decadal basin-scale trends in wind stress throughout the global Ocean[18,19]. Consequently, the global wave climate varies as a response to these atmospheric oscillations[20], and waves change associated with interannual climate variations and teleconnections in both hemispheres, for example the North Atlantic Oscillation (NAO), the El Niño Southern Oscillation (ENSO), or the SAM[7,8,11,12,21]. However, spatiotemporal sea surface temperature (SST) gradients are known to be a critical driver in ocean atmosphere teleconnections and influence wind patterns and storm cyclogenesis globally[22,23]. Furthermore, wave climate projections for the end of the century also indicate that SST warming in the tropical Pacific will lead to increases in wave heights and wave energy levels at mid- to high latitudes over the Southern Ocean and central North Pacific[5,15,24–26], through an intensification of the SAM, ENSO, and NAO[25].

Not only is SST a critical driver in ocean atmosphere circulation but it is also one of the global indicators of ocean warming. The SST was one of the first oceanographic variables recorded from climatic observations in the 1950s and offers the possibility of studying long-term records[27]. Despite these effects of SST on the global wind patterns, SST has been largely overlooked when looking for a global signal of climate change in the historical wave climate.

In contrast with previous studies, we focus on the Wave Power (WP), which measures (over cumulative periods of time) the transport of energy that is transmitted by air-sea exchanges and employed for wave motion[28]. WP has not been studied as a climate change indicator yet, but it can potentially characterize the long-term behavior of the global wave conditions better than wave heights. This article investigates whether or not global wave power (GWP) shows a trend of increase as a consequence of climate change. Furthermore, because the upper ocean has been warming and it is proven that spatiotemporal SST perturbations influence critically wind patterns at a global scale, we examine if there is a direct association between historical SST and WP changes. We use long time series based on historical wind-wave and SST datasets covering from 1948 to 2017. GWP and SST are analyzed based on correlation of the time series and their non-autocorrelated residuals, information theory, long-term trends, and regression analysis. The results show an increasing GWP, and strongly correlated and statistically dependent of SST, globally and by ocean basins. We also find strong inter-regional correlations at the ocean sub-basin level that explain the global increase based on atmospheric teleconnections on winds and waves. These patterns also agree with the predicted future changes in the wave climate. This is the first time that WP has been identified as a potentially valuable climate change indicator. WP can provide new insights into climate change now and in the future.

## Results

**Increasing global WP associated with global warming**. WP is the transport of energy by waves and represents the temporal variations of energy transferred from the atmosphere to the ocean surface motion over cumulative periods of time. WP can be a better indicator of long-term behavior of the global wave conditions than other previously studied parameters because WP (see Methods) includes information on significant wave heights (the mean value of the largest third of the wave heights during typically 1 h, $H_S$) but also periods ($T$), it increases more with the extreme wave heights (being proportional to $H_s^2 T$), and WP can represent accumulated wave energy over periods of time (e.g., months, seasons, and years), unlike other wave climate parameters (e.g., $H_S$) that must be averaged. WP can therefore represent variations in the wave climate not captured by other parameters (e.g., an annual mean wave height).

To investigate trends in WP over time and its relationship with global warming, we calculate and analyze long-term global time series of WP and SST. Changes in wave climate have been previously studied through four types of data: buoy measurements, observations from ships, satellite-based altimetry records, and numerical modeling. This study combines satellite altimetry and model results (validated with buoy measurements) to determine GWP because observations from buoys and satellite altimetry do not provide continuous data over space and time. GWP was calculated using hourly time series of significant wave heights and mean wave periods from four sources (see Methods): the altimetry-corrected global ocean wave (GOW) reanalysis for the period 1948–2008[29]; a global high-resolution hindcast (RaA13) that covers the period 1994–2012 and uses improved parameterizations and high-resolution wind forcing (the Climate Forecast System Reanalysis, CFSR hereafter)[30]; GOW-CFSR, with the same model parameterizations and the wind forcing than RaA13 but with data up to the year 2017[31]; and wave height measurements from satellite altimeter data from 1992–2008, instead of wave modeling results. To calculate mean WP time series, the hourly time series of WP are aggregated by seasons and years and averaged spatially (see Methods). The GWP time series from GOW is highly correlated with the RaA13 (0.69), the

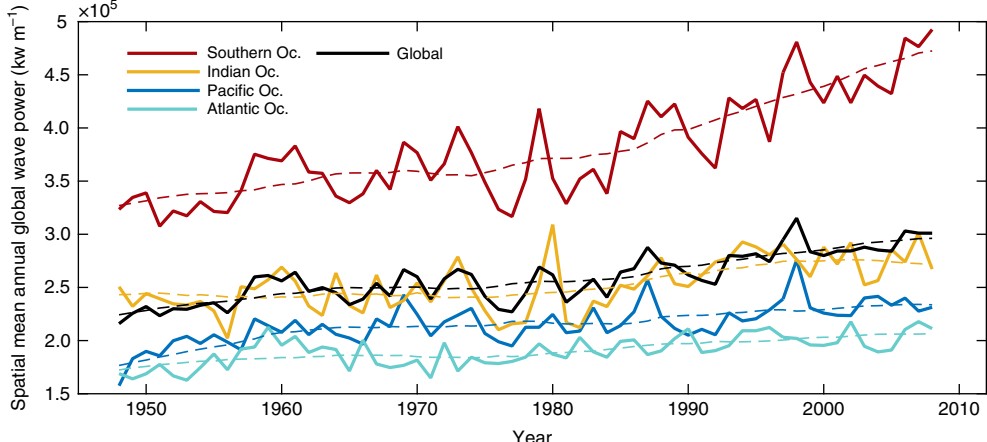

**Fig. 1** Spatial mean annual Wave Power calculated globally and by ocean basin. The dashed lines represent the 10-year moving averages. The Southern Ocean is defined between latitudes of 40°S and 80°S. The mean regional Wave Power is calculated as the spatial average of each historical wave power time series (see Methods). The solid lines indicate each time series. The dashed lines correspond to the 10-year moving average. The time series calculated by latitudinal bands can be found in Supplementary Figure 1

GOW-CFSR (0.86), and the satellite (0.66) GWP time series. The variability in the global time series, as determined by the respective non-autocorrelated residuals (see Methods), are also highly correlated at 0.89, 0.79, and 0.88, respectively. A comparison of the GWP derived from satellite altimetry and the three numerical datasets shows a good agreement (see Supplementary Figure 1).

Based on the long-term time series of WP and SST, we first investigate trends in WP over time to later compare with the SST warming and variability. Figure 1 shows that WP has increased globally by 0.47% change per year from 1948 to 2008 (at 1087 kilowatts m$^{-1}$ year$^{-1}$), and by 2.3% per year since 1994 (GOW-CFSR). The Southern Ocean (defined by the 40°S latitudinal limit, see Methods) is the most energetic basin and dominates the other oceans in terms of WP. It has also increased the most, by 0.58% per year, while the Pacific increased by 0.35% and the Atlantic and Indian Oceans by 0.26% per year. The WP in the Indian Ocean, where local wave generation is small but are subject to swell propagated from the Southern Ocean[11], is close to the average GWP. These trends are statistically significant per the Mann–Kendall approach[32], and the method in Wang and Swail[8] that avoids autocorrelation in the time series (see Methods).

Figure 1 also shows strong interannual variability in the different time series; but more pronounced in the Southern Ocean. This strong variability for the Southern Ocean, also previously found in wave heights, periods, and directions[11], has ramifications in the eastern Pacific and Indian Oceans as seen in the peaks in the Figure during 1980 in the Indian Ocean and 1998 in the Pacific, because these are regions subjected to swell propagated from the Southern Ocean[11,33]. The Supplementary Figure 2 analyzes the trends by latitudinal bands and identifies a recent decline in the WP after the 1990s in the extratropical northern hemisphere, which agrees with decreases in wave heights identified from satellite altimetry in the same region and period[9]. Although the Atlantic Ocean has some of the most extreme wave heights on the planet[8,33,34] (see Supplementary Note 1 and Supplementary Figure 3), its annual WP is the lowest of all ocean basins. This indicates that the WP can represent features of the wave climate not captured by other wave parameters like the mean or extreme values (high percentiles) of wave heights.

Because waves result from the energy transfer of wind to the ocean surface and SSTs influence critically wind patterns

throughout the globe, the increase in GWP could be related to the upper oceanic warming. To examine if GWP keeps some relationship with SST, we use two different SST datasets (see Methods): the most recent version of the extended reconstructed SST (ERSSTv3b) dataset, which provides long-term, monthly SST from 1854 to the present[35]; and the NOAA's optimum interpolation SST (OISST), an independent high-resolution dataset that combines observations from different platforms (satellites, ships and buoys) from late 1981 to the present[36].

The SST anomaly time series shows statistically significant global warming trends. The ERSSTv3b SST has been increasing by 0.06 °C per decade from 1948 to 2008, and the OISST by 0.10 °C per decade after 1981 (Fig. 2b). GWP and SST time series and their variabilities are strongly correlated (Fig. 2a, b). Annual GWP is positively correlated with global ERSSTv3b SST (Fig. 2a, b), with a Pearson's correlation coefficient of 0.86 from 1948–2008 (0.64 during the satellite era 1992–2008), and by 0.56 since 1994 from the GOW-CFSR. The correlation of the GWP with the high-resolution OISST dataset is 0.60 for the period 1948–2008 (0.65 for the satellite era, 1992–2008). The non-autocorrelated residuals (Fig. 2c), which represent the variability in the time series, are also correlated by 0.40 for the same period (0.51 during the satellite era).

GWP and SST are also correlated globally on a seasonal scale (Fig. 3). The contemporaneous (in other words, no time lag) correlation is 0.74 between 1948–2008 (0.64 during the satellite era, 1992–2008). A lagged-correlation analysis shows that the maximum correlation is 0.76, between an SST during a given season and the GWP one season after (time lag = 1; Supplementary Figure 4). Figure 3 also highlights strong El Niño events (annotations overlaid on the Figure) that coincide with spikes in interannual warming and a larger GWP. However, Fig. 3 reveals interannual variations in the global SST anomaly beyond the presence of El Niño events that are probably induced by other patterns.

The statistical dependence of the GWP on SST is studied more deeply using information theory[37]. This technique is applied here because it has been shown useful to determine the influence of SST on hurricane intensity based on analysis of time series in an equivalent application[38]. We calculate the mutual information (MI), which is a measure of the SST information that is shared by GWP and represents the independence between the two variables[37]. If the two variables are independent, GWP contains

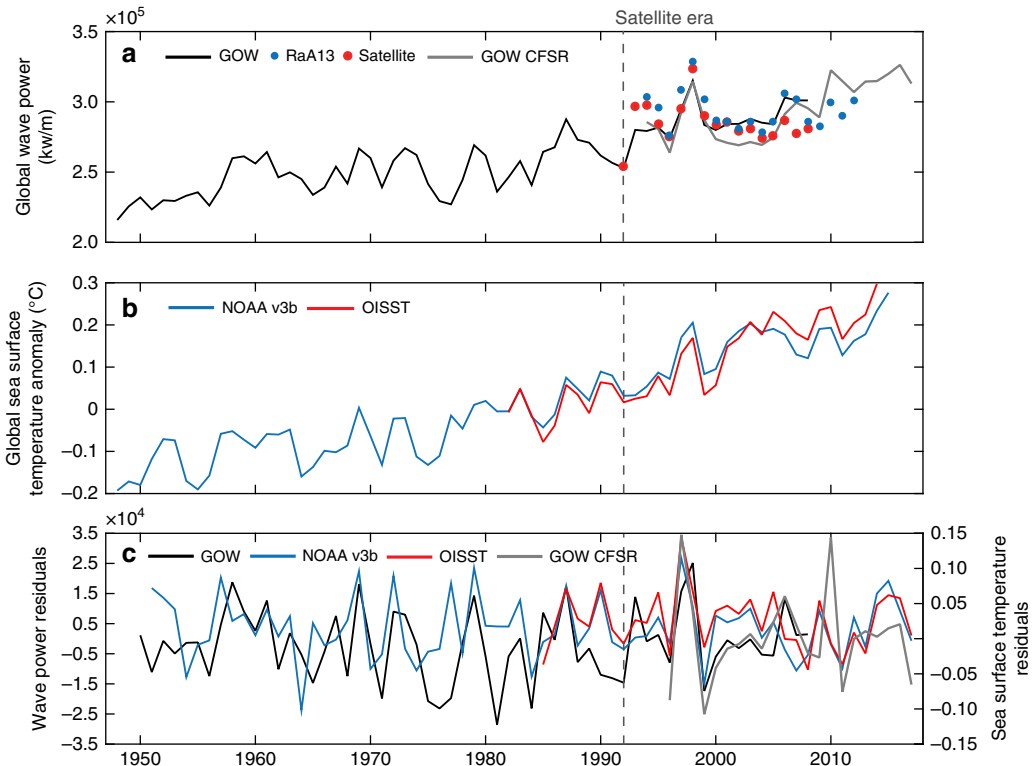

**Fig. 2** Historical variability in oceanographic forcing. **a** Global Wave Power time series from GOW (black: GOW-NCEP, presented in Fig. 1; and gray: GOW-CFSR), RaA13 (blue points) and satellite altimetry (red points); **b** global Sea Surface Temperature time series from ERSSTv3b (blue) and OISST (red); and **c** non-autocorrelated residuals of Wave Power from GOW (black and gray) on left vertical axis) and Sea Surface Temperature from ERSSTv3b (blue) and OISST (red) on right vertical axis. The vertical dashed lines indicate the beginning of the era in which wave height has been measured with satellites

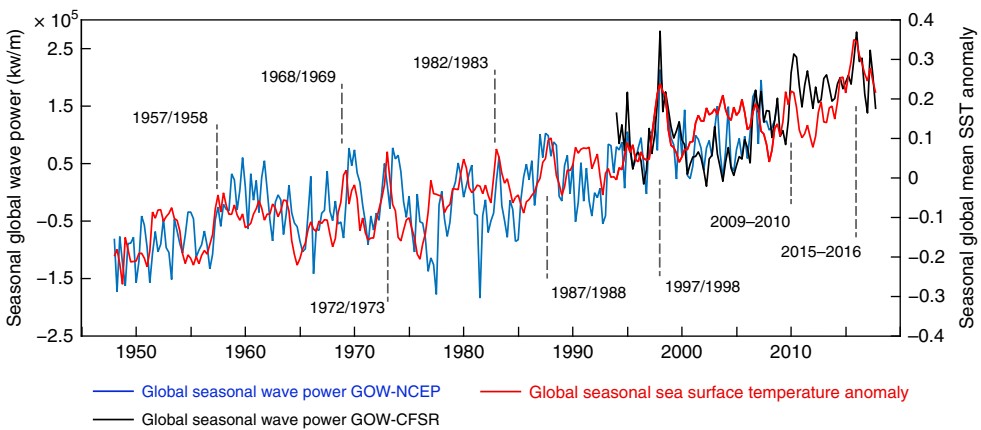

**Fig. 3** Historical seasonal variability in oceanographic forcing Global variability in the seasonal Global Wave Power (blue: GOW-NCEP; black: GOW-CFSR; left vertical axis) and Sea Surface Temperature anomalies (red line; right vertical axis). Years corresponding to strong El Niño events, that is, ones with an Oceanic Niño Index value exceeding 1.5, are annotated and overlaid on the graph

no information about SST and vice versa (i.e., MI = 0; see Methods), and the product of the marginal distributions (Fig. 4c) should replicate the joint distribution (Fig. 4d). However, these distributions differ (Fig. 4c, d), indicating statistical dependence. MI is also 1.5, different from zero, for entropies of 3.1 in GWP and 3.2 in SST. The non-autocorrelated residuals also show dependency (an MI of 1.05 for entropies of 2.86 for the SST residuals and 2.95 for the GWP residuals), indicating that the annual change in WP is also statistically dependent on SST variability on a yearly basis.

GWP has also been increasing linearly with SST at a rate of 1.8 $10^5$ kilowatts m$^{-1}$ year$^{-1}$ over the global ocean (Fig. 5a):

$$GWP \cdot 10^{-5} = 1.804\Delta SST + 2.606 \quad \text{(normalized root mean}$$

square error = 12%). However, not only the time series of the GWP and SST but also the non-autocorrelated residuals (a measure of the signal variability) were also highly correlated (Fig. 2c). We estimate that GWP can change annually by 1.2 $10^4$ kilowatts m$^{-1}$, on average as a result of sea surface warming, for an annual change of 0.1 °C in SST. This represents a ~4.3% annual change in GWP (over the mean GWP), equivalent to a global long-term change over a decade (i.e., 4%, Fig. 1).

**Regional variations in sea surface warming and WP.** Figure 6 shows the spatial trends in WP calculated from 1985–2008 for a

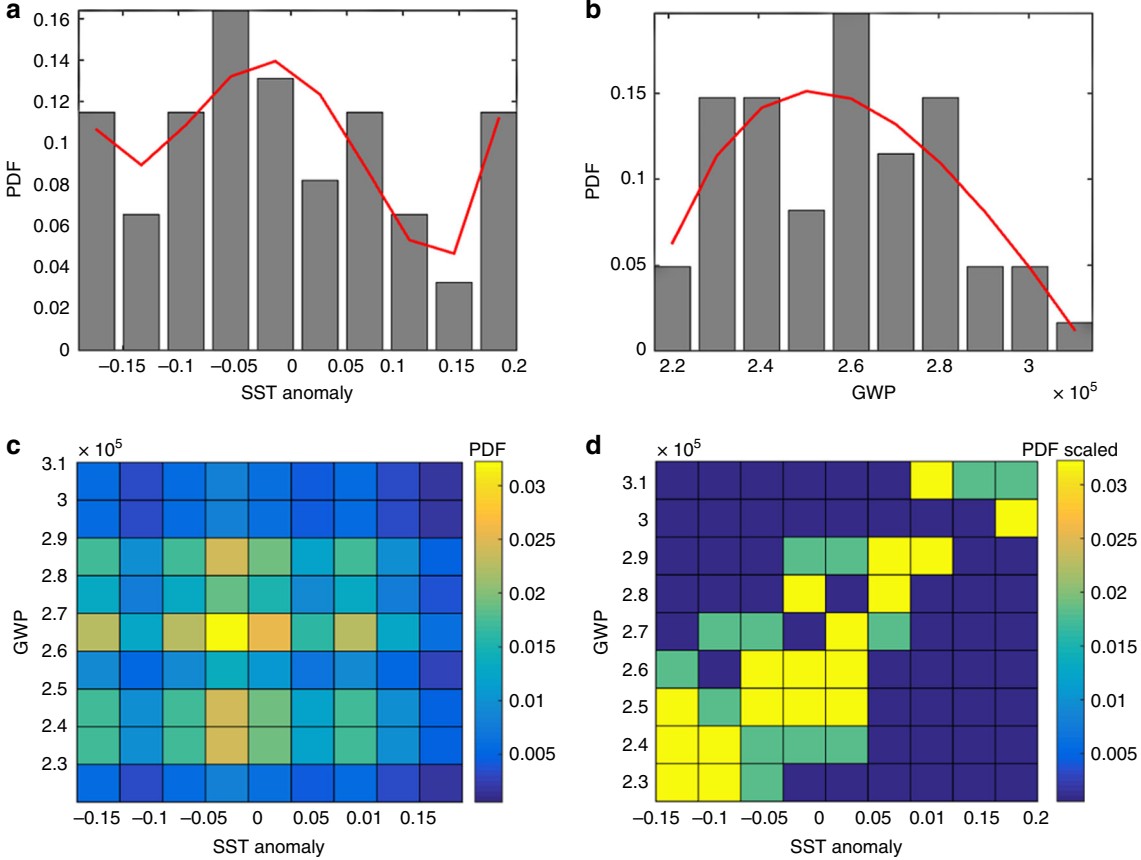

**Fig. 4** Analysis of the statistical dependency of Global Wave Power on the Sea Surface Temperature anomaly. Probability density functions (PDFs) of **a** Sea Surface Temperature (SST) anomaly and **b** Global Wave Power (GWP); **c** the product of their marginal distributions; and **d** their joint distribution (scaled to the values in **c**). The bar plots in **a** and **b** represent the empirical PDF, while the red lines represent a PDF fitted to the data

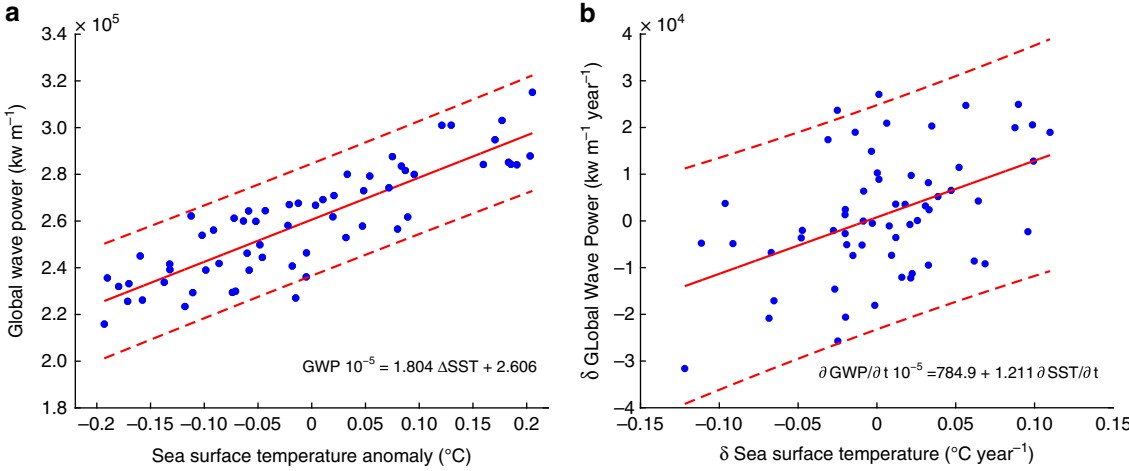

**Fig. 5** Regression between the Sea Surface Temperature anomaly and Global Wave Power. **a** Regression between the global time series of Sea Surface Temperature anomalies (ºC) and Global Wave Power (kw/m); **b** regression between the annual rates of change in Sea Surface Temperature (ºCyear$^{-1}$) and Global Wave Power (kw m$^{-1}$ year$^{-1}$). The solid red lines represent the linear regression lines (equations are noted in each plot). The red dashed lines represent the 95% confidence intervals for each regression line. Both regression lines are statistically significant at the 95% level. The blue dots represent the data for the period 1948–2008

direct comparison with results stablished based on satellite altimetry by Young et al.[9]. The patterns of changes in WP (later discussed in more detail) resemble the patterns in Young et al. for significant wave heights. WP has increased in mid- and southern latitudes by 2.5% per year, with decreases in the North Pacific and North Atlantic between 0.5 and 1% per year. This

recent decrease is also observed in the WP time series for the extratropical northern latitudes (Supplementary Figure 2). The differences in WP calculated in periods of 10 and 20 years demonstrates that WP in the North Pacific and North Atlantic particularly decreased between 1999–2008, when the Pacific Ocean was cooler, but increased during the previous decades

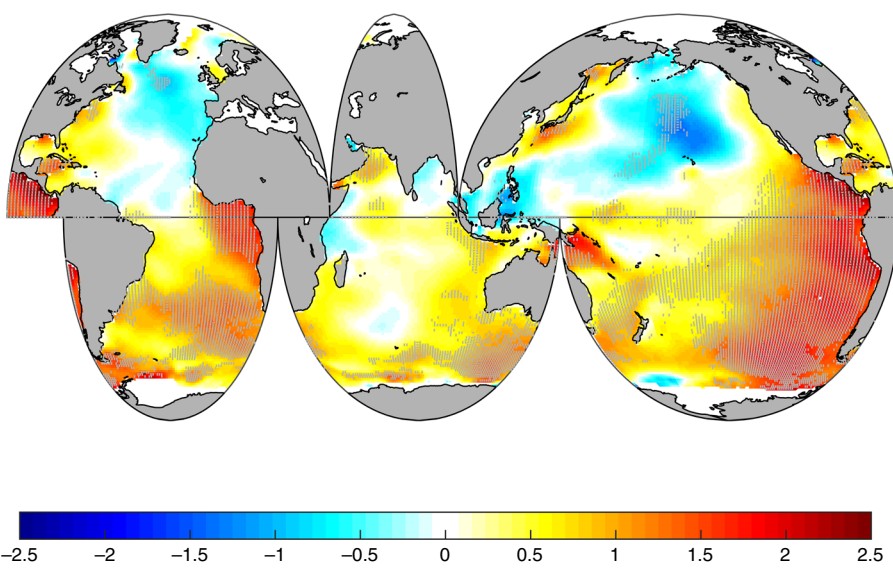

**Fig. 6** Spatial trend (percent change per year) in mean Wave Power from 1985 to 2008. Hatched areas represent points that are statistically significant at the 95% confidence level according to the Mann–Kendall test and the Wang and Swail method for autocorrelation (see Methods). The trends are calculated for the period 1985–2008 (period with satellite-derived wave data) for comparison with[9]. Supplementary Figure 5 shows the spatial trends for other periods in the historical record

(Supplementary Figures 6 and 7). The strong decadal variations in SST and WP suggest that the long-term changes in GWP and SST need to be studied in relation to ocean atmosphere tele-connections and interannual variability before explaining the global connection between the two variables.

Increases in wave height primarily occur from increased surface wind energy[15], but the global wind patterns change in response to spatiotemporal SST variations[22,23]. For this reason, we study spatial SST teleconnections with WP first by calculating the correlation of WP with two SST-based climate indices. Climate indices are diagnostic quantities used to characterize an aspect of a circulation pattern allowing a statistical study of the dependent climatological effects, and have been widely used to explain changes in wind speeds[20], extreme wave heights[8,10,33,39,40], wave direction[11], and wave energy[14,41].

Figure 7 represents the influence of the two most influential SST-based climate indices on WP[14]: Niño3 for the El Niño Southern Oscillation (ENSO) and the Atlantic Multidecadal Oscillation (AMO) index. El Niño (positive ENSO phase) is defined by prolonged, above-average warming of the tropical Pacific (Fig. 7a), and increases cyclonic activity in the Pacific and Atlantic through dynamic atmospheric bridging[42]. During ENSO events, the atmospheric response to SST anomalies in the equatorial Pacific influences ocean conditions globally[43]. ENSO is generally associated with a strengthening of the Aleutian low and high pressures in northwestern America, which direct the wind toward northwestern Canada and Alaska[44]. ENSO events also have a strong influence on WP variability around the globe: during its warm phase, it increases/decreases the North/South Pacific WP (Fig. 7b). This effect can also be observed in the decadal changes in WP and SST (Supplementary Note 2 and Supplementary Figures 6 and 7). Meanwhile, the AMO is a mode of natural variability that occurs in the North Atlantic, with basin-wide SST changes over a period of 60–80 years (Fig. 7c). The AMO index is also associated with increased tropical cyclone activity (as evidenced by the 2017 hurricane season in the North Atlantic, when SST anomalies exceeded + 2 °C off the coast of West Africa). It is correlated with larger WPs in the southern hemisphere and eastern Pacific.

Teleconnections vary over time and space, yet climate indices only represent specific areas of the global Ocean. As an alternative to region-specific indices, we calculate seasonal correlations for the time series and non-autocorrelated residuals of SST anomalies and WP in ocean sub-basins. The selected five sub-basins, defined by the 30°N and 30°S latitudinal limits in each Ocean, are based on oceanic areas used for studying the contribution of ocean swell to the global wind-wave climate[45], which also correspond to different wave climate types[24,46]. More than 90% of the storm-wave maxima is generated in the extratropical sub-basins. The teleconnections of regional SSTs with the WP in these generation areas are particularly useful to explain the increase in GWP from SST warming because, outside them, swells are not significantly affected by surface winds[45] and the local contribution to WP can be considered negligible.

We calculate correlations between the regionally-averaged seasonal time series of SST and WP for the long-term period (1948–2008; Fig. 8a) and the satellite era (1979–2008; Fig. 8b). Table 1 provides the maximum correlation values and the corresponding seasonal lag. Most of these correlations are contemporaneous (that is, in the same season), although some regions exhibit seasonal lags. The regional correlations indicate that SST variations influence the WP in mid- and high latitudes in various basins. The strongest correlations are found between tropical Atlantic SSTs and WP in the extratropical South Pacific and extratropical South Atlantic. The warming in the tropical Pacific (e.g., an El Niño) correlates with increases in the WP in the mid- and extratropical regions of the North Pacific. A closer analysis of the spatial decadal changes in WP and SST in intervals of 20 and 10 years shows the Atlantic and Pacific Oceans were warmer from 1989–2008 than during the previous 20 years (Supplementary Figure 7); particularly, the decade of 1989–1998 was warmer along the eastern-central tropical Pacific, coinciding with a larger WP in the North Atlantic and North Pacific (Supplementary Figure 6). This pattern is consistent with the behavior found for El Nino climate index that represents the ENSO (Fig. 7b). We also find strong inter-regional correlations involving the tropical Indian and Pacific oceans. The tropical Indian Ocean SSTs are highly correlated with extratropical WP in

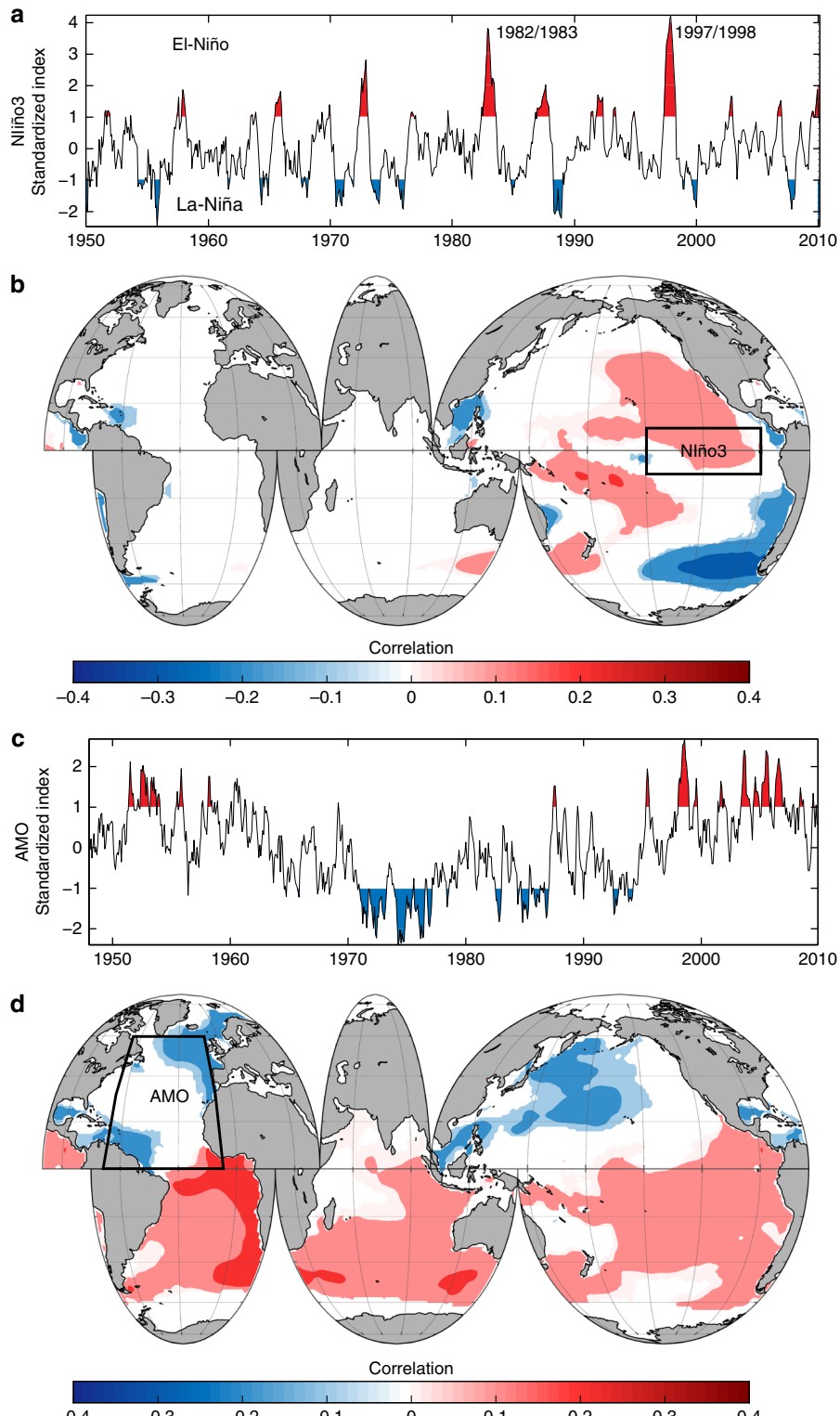

**Fig. 7** Influence of Sea Surface Temperature climate indices on Wave Power. **a** Time series of Niño3 standardized index; **b** spatial correlation pattern of Niño3 with Wave Power; **c** time series of the AMO standardized index; and **d** spatial correlation pattern of the Atlantic Multidecadal Oscillation (AMO) with Wave Power. The Niño3 index registers Sea Surface Temperature anomalies in the tropical Pacific (90°-150ºW, 5ºS-5ºN). The red and blue colored areas in **a** and **c** represent moderate or above events (with absolute value of index of 1 or above) for each corresponding climate index. For the correlation maps **b** and **d**, only the linear correlations that are significant at the 95% level are shown. The polygons represented in the maps in **b** and **d** identify the areas in which each index is calculated based on Sea Surface Temperature anomalies (see Methods)

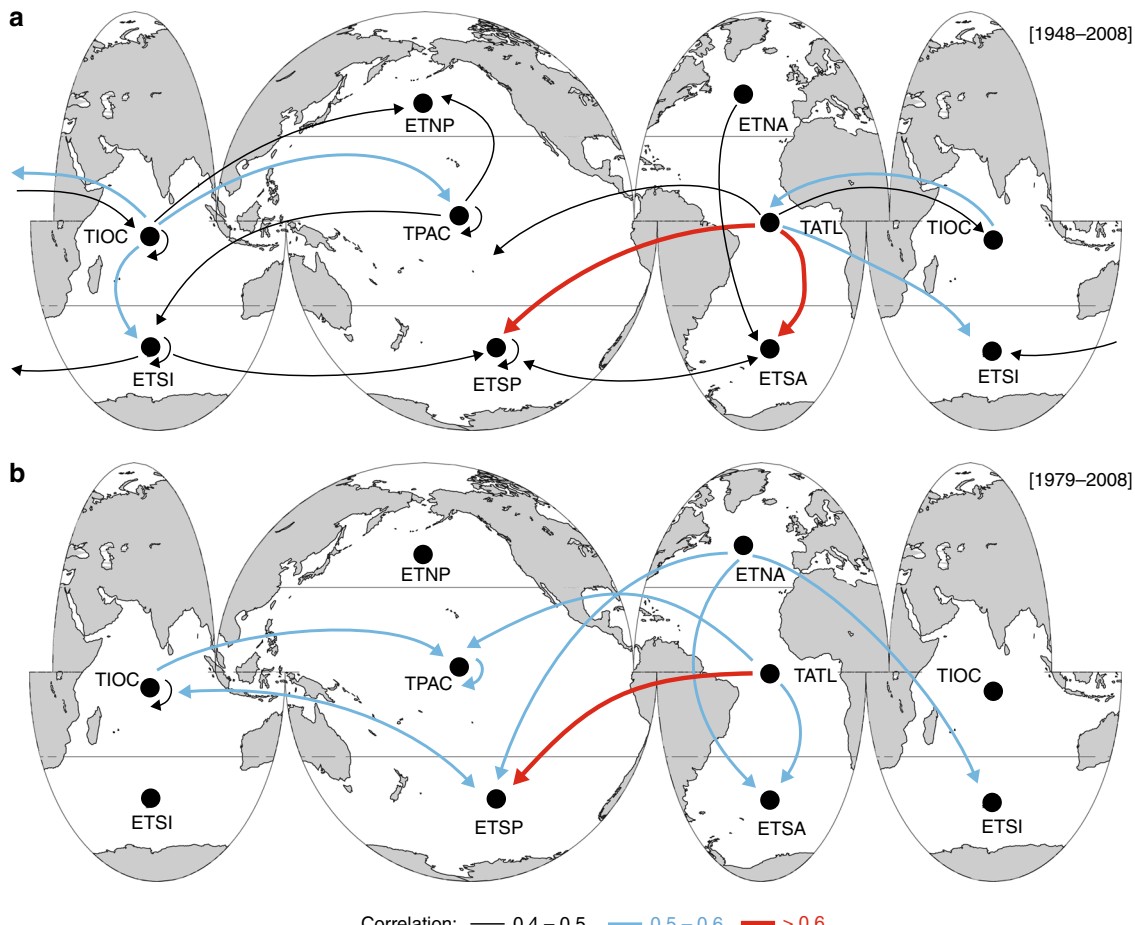

**Fig. 8** Maps of the inter-regional correlations between Sea Surface Temperature and Wave Power. Spatial map of the correlations between the regionally-averaged seasonal time series of Sea Surface Temperature and Wave Power for the periods **a** 1948–2008 and **b** 1979–2008. The arrows size and color indicate the maximum correlation coefficient between the regional Sea Surface Temperature in the origin region (from) with the seasonal Wave Power in the target region (to). Only the correlations that are significant at the 95% level are shown. The correlation coefficients correspond to the maximum values in Table 1. The ocean sub-basins correspond to: extratropical North Pacific (ETNP), tropical Pacific (TPAC), extratropical South Pacific (ETSP), extratropical North Atlantic (ETNA), tropical Atlantic (TATL), extratropical South Atlantic (ETSA), tropical Indian Ocean (TIOC) and extratropical South Indian Ocean (ETSI). Equivalent results for the time series during the satellite era (1979–2008) and the non-autocorrelated residuals can be found in the Supplementary Tables 3 to 6

| Table 1 Inter-regional correlations | | | | | | | | |
|---|---|---|---|---|---|---|---|---|
| | | **WP** | | | | | | |
| | | **ETNP** | **ETNP** | **ETNP** | **ETNP** | **ETNP** | **ETNP** | **ETNP** | **ETNP** |
| SST | ETNP | −0.19 (−2) | 0.16 (0) | 0.25 (0) | — | 0.19 (0) | 0.33 (−1) | 0.22 (−2) | 0.31 (2) |
| | TPAC | 0.48 (0) | 0.48 (0) | 0.53 (3) | 0.23 (−1) | 0.32 (0) | 0.53 (2) | 0.47 (3) | 0.53 (1) |
| | ETSP | 0.41 (−1) | 0.38 (−3) | 0.49 (0) | 0.20 (3) | 0.26 (0) | 0.46 (1) | 0.44 (−3) | 0.45 (−3) |
| | ETNA | — | 0.29 (−1) | 0.38 (0) | — | 0.28 (0) | 0.46 (0) | 0.28 (−1) | 0.44 (−1) |
| | TATL | 0.39 (−2) | 0.48 (0) | 0.66 (1) | 0.21 (−3) | 0.42 (1) | 0.60 (0) | 0.48 (2) | 0.58 (−1) |
| | ETSA | 0.43 (3) | 0.28 (0) | 0.48 (−2) | 0.23 (3) | 0.28 (2) | 0.45 (2) | 0.38 (0) | 0.39 (1) |
| | TIOC | 0.45 (3) | 0.51 (−1) | 0.61 (−2) | 0.26 (−2) | 0.37 (0) | 0.58 (0) | 0.51 (1) | 0.55 (0) |
| | ETSI | 0.40 (−1) | 0.29 (0) | 0.45 (0) | 0.20 (−3) | 0.25 (1) | 0.46 (1) | 0.41 (3) | 0.45 (0) |

Seasonal correlations between average Sea Surface Temperature anomalies in a given sub-basin (rows) and Wave Power in a given sub-basin (columns) calculated for the period 1948–2008. The first value corresponds to Pearson's linear correlation coefficient at the 95% confidence interval, while the second value (in parentheses) gives the time lag for the maximum correlation found. The values in parentheses represent the time lag in terms of the number of seasons. Equivalent numbers for the satellite era and correlations with no time lag can be found in the Supplementary Information
*ETNP* extratropical North Pacific, *TPAC* tropical Pacific, *ETSP* extratropical South Pacific, *ETNA* extratropical North Atlantic, *TATL* tropical Atlantic, *ETSA* extratropical South Atlantic, *TIOC* tropical Indian Ocean, *ETSI* extratropical South Indian Ocean

**Table 2 Autoregressive models. Autoregressive moving-average models parameters used for each dataset**

| Dataset | Number of autoregressive terms | Number of moving-average terms |
|---|---|---|
| GOW-NCEP | 2 | 1 |
| GOW-NCEP seasonal | 2 | 1 |
| Satellite altimetry | 2 | 1 |
| RaA13 | 1 | 0 |
| GOW-CFSR | 2 | 1 |
| ERSSTv3b | 3 | 2 |
| ERSSTv3b seasonal | 1 | 2 |
| OISST | 3 | 2 |

the South Indian, South Pacific, and the tropical Atlantic and Pacific oceans.

## Discussion

From 1948 to 2008, WP increased globally at a rate of 0.41% per year (Fig. 1), but with marked spatial changes by ocean basins (Figs. 1 and 6). Because the increase in wave height is primarily driven by increased surface wind energy[15], we find spatial changes for WP in Fig. 6 that resemble the patterns of increase in the mean and 90th-percentile values of significant wave heights and wind speeds from satellite observations during the same period[9]. The sea surface wind speeds increased from 1988 to 2011, globally, but the increases were stronger in mid-latitudes and during winter[16]. Young et al. found strong changes in mean and high percentiles of wind speed of over 1% per year throughout the southern hemisphere, and larger rates in the tropical Pacific[9]. These patterns in wind speeds produced increases in high percentiles of wave heights of over 1% per year in high latitudes, particularly in the southern hemisphere[9], where we also find the stronger increase in WP, 0.58% per year. Areas of the North Pacific and North Atlantic with weak or no long-term trends in wave heights correspond to weak or negative changes for WP in our results. In these regions, where the contribution of swells from other ocean sub-basins are negligible[45], these results indicate less energy transference from the atmosphere to the ocean surface motion.

Our results agree with the identified spatial trends found for wind speeds and wave heights but are also consistent in magnitude. Considering that WP depends on the square of $H_S$ (i.e. $H_S^2 T$), and that for storm conditions WP accumulates comparatively larger increases than during calm conditions, if we assume a 1% annual change in $H_S$ across the global ocean (the largest trend found by Young et al.), it will correspond to a 2% change per year in WP (for no change in wave period, $T$). Our results show 2% increases per year in many regions of the Southern Ocean (Fig. 6, south of 40 degrees South), for an average 0.58% per year across the basin, which is comparable to the increases in wave heights found by Young et al.

Results show that GWP and SST are closely connected: GWP is strongly correlated with global historical SST from 1948–2008, yearly (Fig. 2) and seasonally (Fig. 3), and it also presents statistical dependency on oceanic warming based on information theory (Fig. 4) and linear regression (Fig. 5). While ocean warming has been greatest near the surface[1], where SST has warmed by 0.06 to 0.1 °C per decade (Fig. 2), climate projections indicate stronger trends in tropical and northern hemisphere subtropical regions in the future[1]. Based on a linear regression between the time series of GWP and SST (Fig. 5a), a 0.06–0.1 °C increase per decade represents a GWP increase between 4.2 and 6.9% per decade (with respect to the mean WP during

1948–2008). Projections of warming in the top 100 meters of the ocean range from ~0.6 °C (Representative Concentration Pathway, or RCP, 2.6) to 2.0 °C (RCP 8.5) by the end of the century, relative to 1986–2005[1]. For these projected increases in SST, the regression gives increases in GWP between 32 to 122% (relative to GWP during 1986–2005; an equivalent change in wave heights would be 16 to 48%, leaving other factors unchanged).

These estimates are comparable to the projected future changes in the wave climate. A multi-model ensemble of wave climate projections show changes in the mean significant wave heights ($H_S$) over 7.1% globally, but occurring predominantly in the Southern Ocean, which is the most energetic basin, with increases of up to 10%[47]. This would represent an increase in WP of 21% (only changes in the mean value of $H_S$). However, projections indeed suggest the high percentiles of $H_S$, which contribute proportionally more to WP (with the square of $H_S$, will increase up to twice as much as the mean values;[5] a behavior consistent with the historical satellite-derived trends found by Young et al.[9]. This will represent a 44% increase in GWP, in the range of our estimates (32–122% relative to 1986–2005, depending on the RCP). However, the foregoing projections estimates correspond to mean wave conditions and were obtained from ensembles of various models from the Intergovernmental Panel on Climate Change Fourth Assessment Report scenarios, which represent lower warming than the RCP8.5 for which we find the most extreme increases (122% for a 2 °C SST warming). Ensemble projections of wave energy flux for the end of the century also show a significant increase (up to 30% at a 100-year return level) for the majority of coastal areas in the southern temperate zone[25]. However, these values only correspond to coastal areas and not the whole basin, and it should be expected that the increase in offshore values in the open Ocean will be larger than in the coastal areas because wave energy is lower closer to the continents[14], also in agreement with our estimates. Future values of GWP based on direct regression with SST are in the order of magnitude of dynamic projections of wave climate, but a direct extrapolation of the results should be taken with caution. Similar semi-empirical relationships have been used to explain variables responses to global warming (e.g., sea level rise[48]), but future GWP will also be determined by climate change effects not captured in the historical data such as future variations in interannual climate patterns, which will be critical in WP variability (see Supplementary Discussion).

The global interplay between SST and GWP is explained through ocean atmosphere teleconnections (see also Supplementary Discussion). The response of WP with SST-based climate indices in Fig. 7 and the SST-WP inter-regional correlation patterns in Fig. 8 align with teleconnections between regional SST changes and storm activity[22,23]. The strongest SST-WP correlation occur between the tropical Atlantic SST and WP in the extratropical South Pacific and South Atlantic and is explained by ocean-atmospheric connections between seasonal fields of sea level pressure, wind, and SST temperature in the tropical Atlantic, eastern Pacific, and Indian oceans[49]. Atlantic SSTs influence remote tropical storm activity in the eastern-central Pacific through a Walker circulation-type response analogous to the ENSO-Atlantic teleconnection[50], and is connected to the AMO (Fig. 7c) and the NAO, which is the most influential pattern on the North Atlantic wave climate[51]. North Atlantic warming drives a NAO-like response[52]: a SST warming in the subpolar and the eastern tropical North Atlantic leads a negative phase of the NAO and less energetic wave climate, consistently with the lack of correlations in Fig. 8 in the mid and extratropical North Atlantic. However, a negative NAO is preceded by an AMO-like SST anomaly, which influences WP in the Southern Ocean, as shown in Fig. 8 in the satellite era (see Supplementary Figures 6 and 7).

Figure 8 also shows that the warming in the tropical Pacific leads to increases in WP in the mid- and extratropical regions of the North Pacific, similarly to the behavior detected in Fig. 7 for El Nino3 index. This pattern is in agreement with previously found effects of ENSO on wave heights and wind speeds based on correlations with El Nino climate indices[16,53]. An increase in tropical SSTs led to large wind speeds in the mid Pacific from 1988 to 2011[16], which explains the local influence of the tropical Pacific SST on WP in Fig. 8 during both the long-term and satellite eras. Furthermore, similar seasonal lags to those in Table 1 have been found before for the effects of ENSO on tropical storm activity where, for example, peaks in ENSO appear in winter in the tropical Pacific but lead to intensified tropical cyclones in the North Pacific two to three seasons later[42]. The tropical Indian Ocean SSTs correlation with the extratropical WP in the South Indian, South Pacific, and the tropical Atlantic and Pacific oceans is explained by an atmospheric bridge[54] that connects SST anomalies in the central equatorial Pacific to those in remote tropical oceans at different temporal scales[55,56].

The smaller number of teleconnections in the satellite era (Fig. 8b) relative to the entire dataset (Fig. 8a) are likely due to a shorter record and the presence of strong interannual variability during those decades, which are dominated by the 1998 El Niño in the central Pacific and the AMO in the Atlantic (Supplementary Figures 6 and 7). Some teleconnections are also likely hidden in the satellite era in the southern hemisphere due to decadal changes associated with the SAM (the principal mode of variability for extratropical atmospheric circulation in the southern hemisphere), which enhances westerlies across the Southern Ocean during its positive phase and increases and rotates wave energy clockwise as the storm belt intensifies[11]. The strong interdecadal variations in WP and SST also reflect the difficulty to find teleconnections and trends over short periods, such as the satellite era, and highlight the need to use long-term records, as in Fig. 8a.

These historical teleconnections also agree with spatial changes projected for various wave parameters for the end of the 21st century. Climate projections for the end of the century show that SST warming in the tropical Pacific will lead to increases in wave height and extreme wave energy levels at mid- to high latitudes over the Southern Ocean and central North Pacific[5,15,24,25]. These changes are triggered by the intensification of the SAM, ENSO, and the NAO[25], with a poleward shift in storm tracks and an enhanced westerly jet[5,6,15,47,57]. The historical patterns in Fig. 8a help explain these effects of SSTs increase on the wave climate, where similar historical SST warming in the tropical Pacific and mid and North Atlantic led to an increase in WP in the same basins (Southern Ocean and the North Pacific) (Figs. 7 and 8).

Understanding how GWP responses to oceanic warming has important implications for coastal adaptation, including anticipating impacts on infrastructure, coastal cities and small islands[2,3]. Because wave action is a key driver of coastal change and flooding, as wave energy increases, its effects can become more profound. Sea level rise will also allow more wave energy to reach shoreward, which will have aggravated consequences. However, regional differences in upper-ocean warming will trigger different WP changes in each ocean basin (e.g., Figure 8). The spatiotemporal variability of these effects is for example apparent by the flooding and erosion impacts in the coastlines of the Pacific during ENSO events[58], which are explained by our WP patterns (Fig. 7b). Regionally, changes in the extratropical generation areas of the Southern Ocean and North Pacific, where the WP is more severe, should receive special attention. A better understanding of teleconnection patterns can also help to anticipate coastal impacts as climate change continues to unfold.

These results indicate that oceanic warming in the different basins has likely led to an increase in GWP through the influence of SST on wind patterns. Consequently, the effect of climate change in oceanic warming has been increasing the global energy transferred from winds to the waves represented in the GWP. The impact of climate warming on the wave climate can therefore be seen in the energy transported by the waves, measured through the GWP as a long-term signal of climate change.

## Methods

**WP data and calculation.** Wind-generated waves modulate air-sea exchanges, dissipating energy, and passing momentum on to ocean turbulence and currents[59]. A fraction of this energy is employed for wave motion[28], which results in a transport of energy by waves, known as WP. For irregular waves, the WP can be obtained from the spectral energy density function, $S(f,\theta)$, where $f$ represents the frequency and $\theta$ represents the direction of waves[60]:

$$WP = \rho g \iint c_g(f,h) \cdot S(f,\theta) df d\theta$$

where $c_g$ represents the group velocity $c_g = \frac{1}{2}\left(1 + \frac{2kh}{\sinh(2kh)}\right)\frac{L}{T}$, $L$ represents the wavelength, $k$ is the wavenumber $2\pi/L$, $h$ represents the water depth, and $T$ represents the wave period. $L$ and $T$ are related through the dispersion equation as follows:

$$\left(\frac{2\pi}{T}\right)^2 = gk\tan h(kh)$$

Sustained conditions of irregular waves (i.e., sea state) are usually described by certain parameters that characterize the spectral shape and can be calculated from it. The wave height associated with the standard deviation of the surface elevation, $H_{m0}$, can be computed from the integral of the spectral density or the order-zero moment, $m_0$, as follows:

$$H_{m0} = 4.004 \cdot \sqrt{m_0}$$

For the swell sea states, $H_{m0}$ is comparable to the significant wave height, $H_S$, which represents the mean value of the largest third of the wave heights in the sea state. Other variables defining a sea state are the peak or mean periods ($T_{02}$ or $T_{01}$, respectively).

The WP for an irregular sea state can be obtained from wave spectral parameters using the following expression[60]:

$$WP = \frac{\rho g^2}{64\pi} \cdot T_e \cdot (H_s)^2$$

where $T_e$ or $T_{-10}$ is known as the energy period. This parameter can be estimated from the spectral shape and other parameters. The mean period, $T_{01}$, and $T_e$ can be related through the following relationship:

$$T_e = \frac{m_{-1}}{m_0} = \alpha T_{01}$$

where $\alpha$ depends on the spectral shape. In this work $\alpha$ is taken to be 0.538, which corresponds to a mean peak-enhancement factor of 3.3 on the JONSWAP spectrum[14]. The assumption of a constant spectral shape introduces some uncertainty into the WP estimates, but the effect is negligible because the error when estimating $\alpha$ is an order of magnitude smaller than that for the effects of $T_{01}$ and $H_S$; furthermore, the periods have less influence than $H_S$ in the WP equation. For the same reasons, correctly modeling $H_S$ is critical for an accurate assessment of the global WP.

Information on wave heights collected using buoys and satellite altimetry do not provide continuous data over space and time, and so require numerical climate reconstructions in order to study historical climate states[61,62]. We use numerical wave models, combined and validated with instrumental sources, to describe the global wave climate across different time periods[14].

The GWP was calculated hourly for time series of significant wave heights ($H_S$) and mean wave periods ($T_S$)[14]. WP is expressed in terms of kilowatts per meter (kw m$^{-1}$) along the wave front. Global wave data containing $H_S$ at hourly intervals, the mean period and the mean directions for the period 1948–2008 were obtained from the GOW reanalysis[29]. The GOW reanalysis used the WAVEWATCH III model[63] with NCEP/NCAR global wind and ice cover datasets[61]. The simulated wave parameters are later calibrated with satellite altimetry from the period 1992–2008 through statistical and directional corrections[64]. Hurricanes and typhoons in the satellite data are identified applying an outlier filter based on weighted least squares at a 95% significance levels and not considered in the calibration[65]. The calibrated data reproduce the spatiotemporal variability of the global wave climate, as demonstrated by the comparison with wave buoys and satellite altimetry, both in terms of wave heights and WP[14,29].

The global long-term wave data from GOW were cross-checked with two other data sources to corroborate the global time series of GWP in terms of magnitude and temporal variability and extend the time series beyond the year 2008. First, we calculated the GWP with wave data from an independent and more recent high-resolution global reanalysis (RaA13) that covers the period 1994–2012 and uses improved parameterizations for the model WAVEWATCH III and high-resolution wind forcing (the Climate Forecast System Reanalysis, CFSR)[30]. RaA13 implemented and validated the improved parameterizations of Ardhuin et al.[66] and found no evidence of bias in wave heights when using unbiased winds after 1994, even for high significant wave height values. Another GWP time series up to the year 2017 is calculated from the global reanalysis GOW-CFSR, which is an updated version of GOW with a resolution of 0.5 degrees globally and uses the model parameterizations in RaA13 and the CFSR wind forcing[31]. Additionally, satellite altimetry data from 1992–2008 (the period coinciding with that of GOW) were used to calculate the significant wave heights via satellite altimetry, and the periods were interpolated in time and space from GOW to provide a third time series of GWP. The altimeter data were produced and distributed by AVISO (http://www.aviso.ocanobs.com/)

However, CFSR winds and waves show large errors over the Southern Ocean until the 1994 compared to altimeter data[30,67]. Chawla et al. explained this high bias by an increase in extreme values of the wind speed[67] but Ardhuin et al. also found that even the mean values are strongly biased for latitudes south of 30°S[30,68], the most energetic region of all the basins (Fig. 1). The high bias is corrected for wave data after 1994 when the CFSR reanalysis started to assimilate wind data from the Special Sensor Microwave Imager (SSM-I)[62]. It is possible that a correction of the CFSR wind speed histogram may be enough to correct the biases on wave parameters, but both the AaR2013 and GOW-CFSR are uncorrected before the year 1994, which induces large overestimation of WP compared to the satellite. Therefore, GOW-CFSR and AaR2013 data are only considered after the year 1994. The GOW-CFSR climatology has been validated with satellite data in the latter period showing a good agreement for mean significant wave heights with some bias in the high latitudes of the southern hemisphere[31]. Additionally, the three numerical datasets (GOW-NCEP, AaR2013, and GOW-CFSR) are compared with the signal of GWP calculated from satellite in the Supplementary Figure 1. The two high-resolution datasets provide a better fit than the long-term GOW-NCEP data, but both present different biases and scatters indices in terms of GWP and are therefore both used in the analysis as independent datasets.

Although numerically-generated wave data are useful to study global wave climatology, there are inhomogeneities caused by changes in the amount of assimilated observations within the forced wind fields throughout the historical time span[69,70]. The two numerical wave datasets used here may present regional discrepancies because they use different wind forcing data and numerical implementations; the GOW uses the NCEP/NCAR reanalysis[61], while RaA13 uses the Climate Forecast System Reanalysis[62] and has a higher spatial resolution (0.5° × 0.5° vs 1.5° × 1.0° in GOW). However, the discrepancies between each wind forcing dataset are minimal in the extratropical storm belts. The climatologies of wind direction and wind speed, as described by the cross-calibration and multiplatform assimilation of ocean surface wind data[71,72], show that wind stress from the NCEP/NCAR reanalysis and climate forcing from the GOW reanalysis only differ from more detailed climate reanalyses in tropical zones (e.g., ERA40 or ERA-Interim). Only in the extratropical storm belts do both wind datasets show similar biases with respect to wind measurements[73,74]. However, the effect of wave generation on tropical zones is negligible in comparison with the effects on high-latitude generation areas[45].

The NCEP/NCAR reanalysis assimilation system[61] was unchanged during the reanalysis period to eliminate climate jumps, although the reanalysis is still affected by changes in the observing data[70,75]. There were two changes in the observing data: before and after 1957 (eras I and II) when the upper-air network was being established and that mainly affected the southern hemisphere;[62] and the introduction of the global operational use of satellite soundings in 1979. It has been therefore recommended that the periods before and after 1979 are studied independently because the climatology based on the years 1979–present day is most reliable[75]. Following this recommendation, our analysis separates the full historical period (1948 onwards) and the satellite era (1979 onwards). The results for the eras not included in the main manuscript can be found in the Supplementary Tables 2 to 6.

**Sea surface temperature**. SST is the water temperature close to the surface of the ocean, usually up to 20 m below the surface of the ocean[76]. SST measurements became more comprehensive and diverse after 1950[27]. Ships and buoys have been recording SST, among many other parameters, for well over 100 years, but after 1967, satellites began to remotely measure SST, with the first global SST composite created in 1970[77]. More widespread satellite measurements of SST since 1982 have allowed for more in-depth exploration of temporal and spatial variations and enabled deeper insights into ocean atmosphere connections.

The SST anomalies used in this analysis were obtained from two sources. We used the ERSSTv3b dataset to define the global signal of upper ocean warming. The ERSSTv3b is the most recent version of the extended reconstructed SST analysis developed by NOAA[35,78]. The dataset is based on the International Comprehensive Ocean-Atmosphere Data Set (ICOADS). ERSST v3b does not use satellite-based SST data but in situ measurements and improved statistical methods to reconstruct temperature fields from sparse data. It provides information on a global 2° grid

from 1854 to the present. The anomalies are expressed in °C and computed with respect to the 1971–2000 monthly climatology[35]. We also use the Optimal Interpolation Sea Surface Temperature (OISST) dataset, which offers higher resolution than ERSSTv3b, providing data on a 0.25° global grid after 1981 and combines observations from different platforms (satellites, ships, buoys)[36]. This dataset is used to validate the ERSSTv3b time series and serve as a second and independent SST dataset to correlate with GWP. The global time series are calculated from monthly SST fields and downloaded from NOAA's National Centers for Environmental Information at: https://www.ncdc.noaa.gov/data-access/marineocean-data/extended-reconstructed-sea-surface-temperature-ersst-v3b; and https://data.nodc.noaa.gov/cgi-bin/iso?id=gov.noaa.ncdc:C00844

**Calculation of the global and regional time series**. The hourly time series of WP were aggregated by seasons and years to calculate the regional and globally averaged time series. The global signals for both WP and SST anomalies were obtained through spatial averaging as follows:

$$Z\text{global} = \frac{\sum_i \cos(\text{latitude}_i) \cdot Z(t)_i}{\text{Number of nodes}}$$

where $Z(t)_i$ represents the annual or seasonal time series of the respective variable at each location in the corresponding grid.

The time series are also calculated by ocean basins and climatic regions. The ocean basins are defined by the continental limits in the Pacific, Atlantic, and Indian Oceans, and the 40 degrees south limit. The Southern Ocean is taken to be south of the 40 degrees latitudinal limit based on WP climatology and storm activity in the southern hemisphere[9,13,14,33]. The regions where we calculated the regional time series corresponded to the wind-wave generation zones and were defined for each ocean sub-basin and a 30° latitudinal limit (see Fig. 8) following Alves (2006)[45]. These oceanic areas have been identified by the propagation footprint of waves[45] and the classification of wave climate types[24,46].

**Calculation of long-term trends**. The long-term trends were calculated for each global time series through a linear regression. The significance of the trends were checked with the Mann–Kendall (MK) test[32]. The purpose of this test was to statistically assess whether or not there was a monotonic upward or downward trend over time by testing if the slope of the estimated linear regression line differed from zero. The MK test is a nonparametric (in other words distribution-free) test and does not require the residuals from the fitted regression line to be normally distributed, as a parametric linear regression does.

Nevertheless, the standard P-values obtained from the MK test are based on the assumption of independence between the observations. This makes it important to check the autocorrelation in a given series and adjust the MK test if necessary. To avoid autocorrelation in the time series, we followed the approach of Wang and Swail[8]. The time series in question is pre-whitened (processed to make it behave statistically like white noise), and the trend is estimated by fitting with the model:

$$Y_t = a + bt + X_t$$

where $a$ and $b$ represent the intercept and slope, respectively, and $X_t$ is given by:

$$X_t = cX_t + \varepsilon_t$$

The pre-whitened time series that possesses the same trend as that of the original time series ($Y_t$) is[8]:

$$W_t = \frac{Y_t - cY_{t-1}}{1 - c}$$

To estimate $W_t$, we follow the iterative approach presented by Wang & Swail[8] as follows:

Step 1. Initial estimate of $c$:

- The lag-1 autocorrelation of the time series $Y_t$ is taken as the first estimate of $c$ (i.e., $c_0$).
- If $c_0$ is less than 0.05, the effect of serial correlation is negligible, and no iteration is necessary when applying the MK test to the original time series.
- Otherwise we perform the described trend analysis on the pre-whitened time series $W_t = \frac{Y_t - c_0 Y_{t-1}}{1 - c_0}$ and obtain the first estimate of $b$ (i.e., $b_0$).

Step 2. Calculation of $b_1$ and $c_1$:

- The estimated trend component from the original data series is removed, and $c$ is re-estimated. The new estimate of $c$, $c_1$, is the lag-1 autocorrelation of the time series ($Y_t - b_0 \cdot t$).
- If $c_1$ is less than 0.05, we take $c_0$ and $b_0$ as the final estimates.
- Otherwise, the trend analysis is performed on the pre-whitened times series $W_t = \frac{Y_t - c_1 Y_{t-1}}{1 - c_1}$, and a new estimate of $b$ is obtained (i.e., $b_1$).

Step 3. Iterations to estimate $b$ and $c$:

- While the differences of abs($c_1$-$c_0$) and abs($b_1$-$b_0$) are larger than 1%, steps 2 and 3 are repeated, which results in $b_0 = b_1$ and $c_0 = c_1$.

- Once the appropriate estimates of $c$ and $b$ are obtained, the MK test is applied to the pre-whitened time series $W_t = \frac{Y_t - cY_{t-1}}{1-c}$ to conclude the trend analysis.

The implementation is accompanied with a Matlab function in the Supplementary Code (see also code availability statement). For the MK test on the pre-whitened series, we adopt the implementation approach of Burkey (2006)[79].

**Relating changes in SST to changes in WP.** Correlations were computed between the global time series, by years and seasons, and spatially over each ocean sub-basin. The correlation was assessed through the Pearson product-moment correlation coefficient, $r$, which is a measure of the strength and direction of the linear relationship between two variables. The statistical significance was calculated through the Student's $t$-test at the 95% confidence level.

Correlations were also calculated for the time series of the non-autocorrelated residuals. The non-autocorrelated residuals were obtained after adjusting the autoregressive moving-average models to each global and regional time series in order to avoid autocorrelation effects in the statistical analysis, and to identify the existence of non-contemporaneous relationships[80]. The non-autocorrelated residuals ignore the effects of trends and autoregression from the original time series and, thus represent the variability (at the temporal scale) of the original time series (yearly or seasonally). Given a time series, $X_t$, we fitted an ARMA(m, n) model by:

$$X_t - (\propto_1 X_{t-1} + \ldots \propto_m X_{t-m}) - (\theta_1 \varepsilon_{t-1} + \ldots \theta_n \varepsilon_{t-n}) = \varepsilon_t$$

where $\propto_m$ is the parameter of the autoregressive part of the model, $\theta_n$ is the parameter of the moving-average part of the model, and $\varepsilon_t$ is the error term. Non-autocorrelation of the residuals was statistically tested with the Ljung–Box test. Table 2 provides the parameters for the adjusted models for each dataset.

Using the global time series of GWP and SST anomalies, linear regressions were calculated between the time series, annually and monthly, as: $GWP(t) = a + b \cdot SST(t - \Delta)$, where $\Delta$ represents a temporal lag, which was found to be zero from a lagged-correlation analysis (Supplementary Figure 3-a).

However, previous studies that use semi-empirical relationships with temperatures to study the effect of global warming on other response variables such as sea level rise[48] and hurricane activity[38,81] have used rates of change, instead of the value of the temperature anomalies, as a descriptor. For this reason, we calculate the yearly changes in the GWP and SST and regress this annual variability as:

$$\frac{\partial GWP(t)}{\partial t} = a + b \frac{\partial SST(t - \Delta)}{\partial t}$$

Each regression model is shown in Fig. 5.

In addition, information theory was used to determine the degree of MI between the WP and SST anomaly. We follow the approach undertaken by Hoyos et al.[38] to study the relationship between SST and hurricane intensity. In information theory, the MI of two variables is quantified to represent the measure of independence of the two variables[37]. MI quantifies the distance between the joint distribution of two variables and the product of their marginal distributions. Therefore, MI measures (in bits) the independence of two variables. MI is based on the concept of entropy, which is associated with the randomness of a variable. The entropy, $H(X)$, of variable X for random event x that occurs with a probability of $f(x)$ is[37]:

$$H(X) = \sum_x f(x) \cdot \log_2 f(x)$$

The joint entropy of two variables, $X$ and $Y$, measures the entropy contained in the joint system and is defined as:

$$H(X, Y) = \sum_{x,y} p(x, y) \cdot \log_2 p(x, y)$$

MI is then defined as:

$$MI(X, Y) = \sum_{x,y} p(x, y) \cdot \log_2 \frac{p(x, y)}{f(x)g(y)} = H(X) + H(Y) - H(X, Y)$$

where $f$ and $g$ represent the marginal distributions of each variable. If $X$ and $Y$ are independent, the total entropy of the system, $H(X,Y)$, would be equal to $H(X) + H(Y)$, in which case the MI value would be zero, indicating that $X$ does not contain information about $Y$, and vice versa. MI was calculated following Hoyos et al. and our implementation was validated against the theoretical cases expressed in the Supplementary Material therein. To quantify MI, we estimate the marginal distribution of the variables (Fig. 4a, b). The product of their marginal distributions (Fig. 4c) should replicate their joint distribution (Fig. 4d) if the two variables are independent, in other words, if GWP contains no information about SST and vice versa (i.e., MI = 0). However, the distributions in Fig. 4c, d differ, which implies that there is statistical dependence between the variables.

Nevertheless, the analysis of the original GWP and SSTs time series does not enable us to directly discern whether or not these statistical relationships arise from

long-term trends or short-term modes of variability (in other words, on a decadal or shorter timescale). To remedy this, we performed the MI analysis on the isolated trend/variability time series following Hoyos et al.[38]. The trend is removed by subtracting the least-squares linear fits and the adjusted autoregressive moving-average models to calculate the non-autocorrelated residuals. The resulting non-autocorrelated residuals also have an MI different from zero (1.05 bits for entropies of 2.86 in the SST residuals and 2.95 in the WP residuals), indicating that WP is statistically dependent on SST variability.

**Climate indices.** Climate index data for Nino3 and AMO for Fig. 7 were obtained from NOAA. The AMO is a mode of natural variability that occurs in the North Atlantic Ocean, with basin-wide SST changes over a period of 60–80 years[82]. The Niño3 index[83] registers SST anomalies in the tropical Pacific (90–150ºW, 5ºS-5ºN). Historically strong El Niño events were defined based on the Oceanic El Niño Index after 1950, which uses the same region as the Niño 3.4 index: (5ºN-5ºS, 170°-120ºW). The Oceanic El Niño Index uses a 3-month running mean; to classify events as full-fledged El Niño or La-Niña occurrences, the anomalies must exceed + 0.5 °C or -0.5 °C for at least five consecutive months. This is the operational definition used by NOAA. The time series of each climate index (Fig. 7a, c) were correlated with each local time series of WP from GOW in the global grid to provide the patterns shown in Fig. 7b, d.

**Code availability.** The statistical significance test for trend calculation is described and provided in a Supplementary File and can be found at: https://osf.io/pvm6c/, with the identifier https://doi.org/10.17605/OSF.IO/PVM6C. Other code used to generate the results for this project are available from the corresponding author upon reasonable request.

## Data availability

The Global Wave Power data from GOW-NCEP and GOW-CFSR that support the findings of this study are available at PANGEA, with the identifier https://doi.pangaea.de/10.1594/PANGAEA.896536. Other data that support the findings of this study are available from the indicated sources or from the corresponding author upon reasonable request.

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

## Acknowledgements

We thank the organizations that provided the data for the analysis in this study: the National Oceanic and Atmospheric Administration (NOAA) and Rascle, N. & Ardhuin, F.; and the helpful insight and comments from Roberto Minguez (autoregression analysis) and Curt Storlazzi. B.G.R. is also indebted to The Nature Conservancy for support provided during the development of this study. Funding for this project was partly provided by RISKOADAPT (BIA2017-89401-R) Spanish Ministry of Science, Innovation and Universities and the ECLISEA project part of the Horizon 2020 ERANET ERA4CS (European Research Area for Climate Services) 2016 Call.

## Author contributions

B.G.R., I.J.L. and F.J.M. conceived and designed the analysis. B.G.R. conducted the analysis. B.G.R., I.J.L. and F.J.M. jointly wrote the manuscript.

## Additional information

**Competing interests:** The authors declare no competing interests.

