## [Peer Review File · Nature Communications]

Reviewer #1 (Remarks to the Author):

NCOMMS-17-30756

Advice: Invite the author to revise their manuscript to address specific concerns before a final decision is reached.

Major comments:

Figure 1A: Both GOW and GOW-SAT appear in the legend. What does GOW-SAT stand for, the GOW data that have been corrected using the SAT data? If yes, were the corrections applied to the data only for satellite era (which seems to be indicated in this figure)? It would be problematic if the corrections were applied to only the satellite era, rather than the whole period of GOW data, because such partial correction could introduce temporal inhomogeneity to the data time series shown here. The corrections should have been applied to the entire record of the GOW data.

Other comments:

1. I think the writing of the manuscript has large room for improvement. Some parts of the text, e.g., the caption of Figure 5, is hardly understandable. I guess the caption of Figure 5 should be something like "Probability density function (pdf) of (A) SST anomaly, (B) global WP, (C) product of the marginal distributions, and (D) the joint distribution scaled to the values in (C)." This manuscript needs substantial editing.
2. L40-41 (Lines 40-41): Wang et al. (2008) detected external influence on trends of ocean wave heights, which I think should be mentioned here at least.
3. L43: Many more studies can be cited here. At least some of the studies listed in the References below should be cited here.
4. L62: This sentence does not seem to fit in the context here.
5. L79 and L474: I think the Mann-Kendall (MK) approach does not take into account the effect of auto-correlation, while the MK method of Wang and Swail (2001) does and has been found to perform best in comparison with other trend calculation methods (IPCC AR5 WG I Report, page 2SM-12, Table 1.SM.3).
6. L85: Please expand "non-autocorrelated residuals" to "non-autocorrelated residuals (see Methods)".
7. Fig. 1A: What does the GWP trend pattern look like? Is an increasing GWP seen everywhere over the global oceans? If not, I think such trend pattern is worth presenting, at least in the Supplementary Material, and regional mean WP time series should be shown for each region of the same trend direction.
8. L106: Correlation is between two time series. Or are you talking about autocorrelation? It is inappropriate to say "The correlation of each time series is also significant..." Maybe "The correlation of each pair of the time series is also significant..."?
9. L108-109: I would modified the sentence to "... analysis shows that the maximum correlation is 0.76 and occurs between SST in a season and the GWP one season after (time lag =1)."
10. L137-138 and L142-144: Kumar et al. (2016) also studied global impacts of natural climate variability such as ENSO, NAO, and PDO on extreme significant wave height, which is worth citation here. Wang and Swail (2001) also related NAO to wave extremes.
11. Fig. 3: What does the shading scheme represent, correlations?
12. L169: Please add a reference for the statement "Outside their generation zone, swells are not significantly affected by surface winds."
13. L214-215: Many more studies on projected future changes in wave climate can be cited here. At

least some of the studies listed in the References below should be cited here.

14. L215-216: Will all SST variations (both increase and decrease) lead to increased wave height? I think you meant to say "SST changes in the ... will lead to wave height changes over ...".

15. L217-219: Did you analyze this? or Ref. 48 did this? Please clarify.

16. Table 1 and elsewhere: I think it is less confusing to use lagged-correlation in place of cross-correlation, because the latter is usually used for the correlation between two time series.

17. L419: "to remove"? Maybe "to set aside"?

18. L435: "...in each signal" \diamond "...in each data time series".

19. Table 2, column header: "Autoregressive terms" \diamond "Number of autoregressive terms". "Moving Average terms" \diamond "Number of moving average terms".

20. L511: Change to "...randomness in a variable. The entropy, $H(X)$, for the variable X of ..."

21. L516: "since $H(X)$ is only equal to $H(X,Y)$ " should be "since $H(X,Y) = 0$ in this case."

22. L522: "Figure 5C and d" \diamond "Figures 5C and 5D".

Supplementary Material:

L11-12: Can cite Wang et al. 2013 and Wang et al. 2006 here (about inhomogeneities in several reanalysis datasets).

L14: Can cite Wang et al. 2006 here (about inhomogeneity of NCEP reanalysis)

L90: Kumar et al. (2016) also studied global impacts of natural climate variability such as ENSO, NAO, and PDO on extreme significant wave height, which is worth citation here. Wang and Swail (2001) also related NAO to wave extremes.

References:

Caires, S., V. R. Swail, and X. L. Wang, 2006: Projection and analysis of extreme wave climate. *J. Climate*, 19 (No. 21), 5581–5605.

Kumar, P., S.-K. Min, E. Weller, H. Lee, and X. L. Wang, 2016: Influence of Climate Variability on Extreme Ocean Surface Wave Heights Assessed From ERA-Interim and ERA-20C Reanalyses. *J. Clim.*, 29, 4031-4046.

Wang, X. L., Y. Feng, V. R. Swail, and A. Cox, 2015: Historical trends in the Beaufort-Chukchi-Bering Seas surface winds and wave heights, 1970-2013. *J. Clim.*, 28, 7457-7569.

Wang, X. L., Y. Feng, and V. R. Swail, 2015: Climate change signal and uncertainty in CMIP5-based projections of global ocean surface wave heights. *JGR-Oceans*, 120, 3859–3871.

Wang, X. L., Y. Feng, and V. R. Swail, 2014: Changes in global ocean wave heights as projected using multi-model CMIP5 simulations, *Geophys. Res. Lett.*, 41, 1026-1034.

Wang, X.L., Y. Feng, G. P. Compo, V. R. Swail, F. W. Zwiers, R. J. Allan, and P. D. Sardesमुख, 2013: Trends and low frequency variability of extra-tropical cyclone activity in the ensemble of Twentieth Century Reanalysis. *Clim. Dyn.*, 40, 2775-2800.

Wang, X.L., Y. Feng, and V. R. Swail, 2012: North Atlantic wave height trends as reconstructed from the twentieth century reanalysis. *Geophys. Res. Lett.*, 39, L 18705.

Wang, X. L., V. R. Swail, F. W. Zwiers, X. Zhang, and Y. Feng, 2008: Detection of External Influence on Trends of Atmospheric Storminess and Ocean Wave Heights. *Climate Dynamics*, 32, 189-203.

Wang X. L. and V. R. Swail, 2006: Climate change signal and uncertainty in projections of ocean wave heights. *Climate Dynamics*, 26, 109-126.

Wang, X. L., V. R. Swail and F. W. Zwiers, 2006: Climatology and changes of extra-tropical cyclone activity: Comparison of ERA-40 with NCEP/NCAR Reanalysis for 1958-2001. *J. Climate*, 19, 3145–3166.

Wang, X. L. and V. R. Swail, 2002: Trends of Atlantic wave extremes as simulated in a 40-year wave hindcast using kinematically reanalyzed wind fields. *J. Clim.*, 15, 1020-1035.

Wang, X. L. and V. R. Swail, 2001: Changes of Extreme Wave Heights in Northern Hemisphere Oceans and Related Atmospheric Circulation Regimes. *J. Clim.*, 14, 2204-2221.

Reviewer #2 (Remarks to the Author):

I have just finished reviewing the manuscript 'Uncovering a new global signal of climate change: evidences of a stronger global wave climate associated to ocean warming'. The manuscript deals with the interplay between sea surface temperature and wave power, a very interesting topic per se. The research gains additional interest given (i) the (large) spatial and (long) temporal scales; and (ii) the fact that it resolves teleconnections among ocean basins. Finally, the paper is scientifically robust and based on sound datasets and methodologies and interesting discussion of the output.

Given all the above I read the manuscript with pleasure, however I have some issues about the presentation, therefore I cannot recommend for direct publication.

My main issue is that the paper does not appear to be prepared for Nature Communications and I find that the authors could improve a lot the manuscript to that direction. For the moment the manuscript is incomprehensible without the methods and the SI and the reader needs to do several iterations along of each of them to gain insight on the work. I think this could be improved substantially.

The authors could dedicate a paragraph after the introduction providing an overview of the methodology.

Phrases like 'as applied in 12' found in line 117 could be the way to go for other journals, but my opinion is that there are more appropriate ways to present the work in the present journal; which would allow the reader to get the main message without having to read several other papers. Also references are in general used like in the [author, year] format and I think for Nature style (number) they should incorporated in a different fashion in the text. I am also not sure but I have the feeling that Nature communications has a limit on the max number of references.

When the authors discuss the results they could include only the relevant information. For example, already from line 48 it becomes confusing for the reader to follow the results with the trends from the different datasets. In my opinion discussing the trends from different datasets should not be the point, but the authors should focus in the most reliable information. The same applies for the SST datasets.

The presentation of the MI results was another aspect that forced me to go through the methods and the manuscript several times. First of all, it is not clear why the authors selected this method to test dependence among the two variables, since there are other more common and simpler to interpret statistical methods. The authors could elaborate a bit on that in their reply.

Figure 3 is aesthetically good, and tries to provide some basic concepts, but is a bit confusing. In the upper row there are patches for values beyond -1, 1 but it is not explained why. The lower map plots have color variations which are not explained by any colorbar. Moreover, it is not clear whether the spatial extent of the colored areas refers to a specific year, or some mean behavior. Moreover, the authors mention spatial influence of the indices to WP, but no evidence is shown. So in this format it doesn't really help a lot.

Figure 4 together with Table 1 summarizes the main findings of the work and give a good insight in the teleconnection patterns. There are differences between the two datasets and the authors provide some explanations, but still some aspects have not been clarified. Could the fewer teleconnections in the satellite era be due to the shorter dataset?

The next important point the authors make is the correlation between SST and WP. First of all this I

find confusing that two equations are used for the correlation. I also believe that Figure 1 of SI should be included in the manuscript. Apparently such a regression fit would have severe implications under global warming, however such an increase has not been found by studies dedicated to wave projections. Could be a limitation of GCMs, or other factors come to play and impede the direct effect of SST on WP? If that would be the case shouldn't there be an increasing trend in WP at least for the shorter future? Discussing more in detail the possible reasons for that discrepancy between present findings and future trends would be very important, as this is a key point.

A less important point... It is confusing that in the several tables discussing correlations between SST and WP the word autocorr appears. Is this correct, since we are referring to two variables?

Reviewer #3 (Remarks to the Author):

Review of the NCOMMS manuscript "Uncovering a new global signal of climate change: evidences of a stronger global wave climate associated to ocean warming" by Reguero et al.

Recommendation:

Reject

Fundamentals:

The paper presents a study where the supposed increase in global wave energy flux is related to the recent increase in SST. The study is base in the GOW wave data set and on a NOAA global SST composite data set.

Relating SST to wave fields, directly, has a certain degree of novelty (some other authors have done it for other parameters, through). Nevertheless the manuscript has some flaws, lacking depth, that prevent it, in my opinion, to be accepted in such a high level journal. The text is not uniform (clearly it has been written by several hands).

Comments:

L22: variables or parameters?

L22-23: what do you mean with "One for the global wave climate has not been found"? Several studies have looked at the impact of present climate change and projected changes in climate in the global wave field (on Tm, Hs, and some even on global wave energy flux), hence how can you state this?

L25: could? Here and in several parts of the text you use "could". It seems you don't have confidence in your results. Why could and not can?

L28-28: Are you sure that you are the first to present the global wave energy flux as an indicator of climate change? And is it an indicator or a consequence of climate change?

L37: what are remote sensing of changes? This whole sentence (L36-39 is confusing).

L39: why broad?

L40: likely? See comment above regarding L25.

L44: Satellites are just platforms. They measure nothing. Remote sensing measurements would be the correct statement. What about the wave heights in the last 9 years (from 2009 to 2017)? Why don't you include these last 9 years?

L48-49: Confusing.

L56: What has been used to investigate surface winds?

L59-60: How can you state that "This may be an indication of a direct correlation between ocean surface warming and wave climate changes, but such a global connection remains to be determined", right having stated two references (14 and 15) that do that? I don't understand what is new here.

L62: Again could? And how are changes in SST related to storm patterns? References?

L72: What do you mean with "WP is cumulative" in comparison with other wave parameters?

L74: ship observation or visual observations from VOS?

L75: Reference 22 is wrong in this context.

L85: Are 0.66 and 0.69 "high" correlations?

L128: energy or speed?

L129-130: "Long term changes and interannual patterns are closely associated". Isn't this a truism? Are they the same in this context?

L135: What are "known relationships" in this context?

L141: AMO is not defined.

L156-164: I don't know if this is a practice of Nature, but stating conclusions and interpretation on figures captions is not exactly very correct.

L169: wave swell or just wave generation areas?

L168: "Outside their generation zone, swells are not significantly affected by surface winds.". And? What do you exactly want to convey here?

L182: TATL is defined on figure's 4 caption, and should be in the text body.

L193: warming SST or increasing SST?

L197: larger or higher?

L206: many is not exactly a very correct dimension... Several, maybe.

L210: storm belts? What is this?

L247: Why 5%? Where is this shown? Isn't this speculative?

L253: 0.1°C increase leads to 6.4% increase in the global wave energy flux... Therefore 2.0° in the RCP8.5 will lead to an increase of...?

L394: significant wave height.

L384: Statement starting with "For swell..." is not correct. What are swell dominated sea states?

L415: Why only until 2008?

L416: Why didn't you use the GOW2 data set?

L428-429: Did you use Tm from altimetry measurements?

L438: "... different wind forcing data"... which are...?

L444: ERA40 is not a reference to be stated. There are much more accurate products (starting with ERA-Interim).

Point by point responses to referees

Reviewers' comments:

Reviewer 1

(Remarks to the Author):
NCOMMS-17-30756

Advice: Invite the author to revise their manuscript to address specific concerns before a final decision is reached.

Major comments:

Figure 1A: Both GOW and GOW-SAT appear in the legend. What does GOW-SAT stand for, the GOW data that have been corrected using the SAT data? If yes, were the corrections applied to the data only for satellite era (which seems to be indicated in this figure)? It would be problematic if the corrections were applied to only the satellite era, rather than the whole period of GOW data, because such partial correction could introduce temporal inhomogeneity to the data time series shown here. The corrections should have been applied to the entire record of the GOW data.

Authors:

Former Figure 1 (now Figure 2) has been substantially edited and revised. In the original version, GOW-SAT was referring to the subset of GOW data for the satellite era (1992-2008). However, this nomenclature was misleading. In the revised version, we use a vertical dashed line to indicate the start of the satellite era instead.

However, we would like to clarify that the GOW dataset does include the satellite-derived corrections for the whole period of GOW data (i.e. homogeneously, see Reguero et al., 2012), as the reviewer correctly points out. This has also been better described in the text (Lines 123-129).

Other comments:

1. I think the writing of the manuscript has large room for improvement. Some parts of the text, e.g., the caption of Figure 5, is hardly understandable. I guess the caption of Figure 5 should be something like "Probability density function (pdf) of (A) SST anomaly, (B) global WP, (C) product of the marginal distributions, and (D) the joint distribution scaled to the values in (C)." This manuscript needs substantial editing.

Authors:

The writing and organization of the paper has been considerably revised and modified (the language editing service, as suggested by the Editor, has greatly helped in this regard). The Figures and captions have also been revised and substantially improved.

The main changes include: substantial language revision throughout the manuscript, changes in Figures and captions, reorganization of sections and ideas to facilitate the overall flow and the narrative.

With regard to the former Figure 5 caption (now Figure 4, in the Results section) now reads:

Figure 4. Analysis of the statistical dependency of WP on the SST anomaly. Probability density functions (PDFs) of (a) SST anomaly and (b) GWP; (c) the product of their marginal distributions and (d) their joint distribution (scaled to the values in c). The bar plots in a and b represent the empirical PDF, while the red lines represent a PDF fitted to the data.

We also note that the explanation we give on the analysis and method has also been improved. Now it facilitates the understanding of the Figure and the associated methods – Lines 183-197.

2. L40-41 (Lines 40-41): Wang et al. (2008) detected external influence on trends of ocean wave heights, which I think should be mentioned here at least.

Authors:

Agree, the reference has been added.

3. L43: Many more studies can be cited here. At least some of the studies listed in the References below should be cited here.

Authors:

Although we note we have a limitation in the number of references, we agree these citations are critical to provide context for the analysis. We have decided to delete other references and add more references in this section, but also in other relevant parts of the manuscript, as suggested by the reviewer.

As a guide, the journal suggests 70 references. However, we share the reviewer's point and we will suggest leaving these additional references as they are important for contextualizing this work.

4. L62: This sentence does not seem to fit in the context here.

Authors:

The transition between paragraphs has been revised and corrected to improve the flow and the readability. The text where this sentence was originally has been substantially revised, along with the overall narrative of the Introduction, to address the reviewer's concern.

5. L79 and L474: I think the Mann-Kendall (MK) approach does not take into account the effect of autocorrelation, while the MK method of Wang and Swail (2001) does and has been found to perform best in comparison with other trend calculation methods (IPCC AR5 WG I Report, page 2SM-12, Table 1.SM.3) .

Authors:

The reviewer is correct, the result of the Mann– Kendall test depends strongly on the autocorrelation. As stated in Wang and Swail (2001) – Annex A: *“If there is a positive autocorrelation in the time series, the test rejects the null hypothesis H_0 more often than specified by the significance level (von Storch and Navarra 1995).”*

We have followed the description in (Wang & Swail 2001) (WS2001), and implemented the iterative method following the paper indications and page 2SM-12 in the IPCC AR5 WG I Report. Our implementation is included in a Matlab function and now provided in an annex in the revised Supplementary Material. The Mann-Kendall test on the prewhitened series follows the implementation in Burkey (2006). The implementation is outlined at the end of this response.

The trends are significant both for SST and GWP using the WS2001 approach.

To validate the results in case we had committed any error in our implementation, we repeated the tests also with an existing R package, i.e. zyp (<https://cran.r-project.org/web/packages/zyp/zyp.pdf>). This R routine computes a prewhitened nonlinear trend on a vector of data, using WS2001 method of prewhitening, the Sen's slope and the Kendall test for significance. With the independent R package, the results were similar: trends were significant at the 95% confidence.

We have modified the text in the main manuscript and in the methods accordingly. The methods include a description of the approach – Line 538 onwards:

“Nevertheless, the standard P-values obtained from the MK test are based on the assumption of independence between the observations. This makes it important to check the autocorrelation in a given series and adjust the MK test if necessary. To avoid autocorrelation in the time series, we followed the approach of Wang and Swail (Wang & Swail 2001). The time series in question is pre-whitened (processed to make it behave statistically like white noise), and the trend is estimated by fitting with the model:

$$Y_t = a + bt + X_t$$

where a and b represent the intercept and slope, respectively, and X_t is given by:

$$X_t = cX_{t-1} + \varepsilon_t$$

The pre-whitened time series that possesses the same trend as that of the original time series (Y_t) is (Wang & Swail 2001):

$$W_t = \frac{Y_t - cY_{t-1}}{1 - c}$$

To estimate W_t , we follow the iterative approach presented in Wang & Swail (2002) as follows:

Step 1. Initial estimate of c :

- The lag-1 autocorrelation of the time series Y_t is taken as the first estimate of c (i.e., c_0).
- If c_0 is less than 0.05, the effect of serial correlation is negligible, and no iteration is necessary when applying the MK test to the original time series.
- Otherwise we perform the described trend analysis on the pre-whitened time series $W_t = \frac{Y_t - c_0 Y_{t-1}}{1 - c_0}$ and obtain the first estimate of b (i.e., b_0).

Step 2. Calculation of b_1 and c_1 :

- The estimated trend component from the original data series is removed, and c is re-estimated. The new estimate of c , c_1 , is the lag-1 autocorrelation of the time series $(Y_t - b_0 * t)$.
- If c_1 is less than 0.05, we take c_0 and b_0 as the final estimates.
- Otherwise, the trend analysis is performed on the pre-whitened times series $W_t = \frac{Y_t - c_1 Y_{t-1}}{1 - c_1}$, and a new estimate of b is obtained (i.e., b_1).

Step 3. Iterations to estimate b and c :

- While the differences of $abs(c_1 - c_0)$ and $abs(b_1 - b_0)$ are larger than 1%, steps 2 and 3 are repeated, which results in $b_0 = b_1$ and $c_0 = c_1$.
- Once the appropriate estimates of c and b are obtained, the MK test is applied to the pre-whitened time series $W_t = \frac{Y_t - c Y_{t-1}}{1 - c}$ to conclude the trend analysis.

The implementation is accompanied with a Matlab function in the Supplementary Material. For the MK test on the pre-whitened series, we adopt the implementation approach of Burkey (2006) (Burkey 2006).

“

The approach of WS2001 is described in an Annex in the Supplementary Material and implemented as follows:

```
function [taub tau h sig Z S sigma sen n senplot Cllower Clupper D Dall C3] =
WS2001_ktaub(t,Yt,alpha,wantplot)

% STEP 1)
c0 = autocorr(Yt); c0 = c0(2);
t1 = t(2:end);

if c0<0.05
    [taub tau h sig Z S sigma sen n senplot Cllower Clupper D Dall C3] = ktaub([t(:) Yt(:)], alpha, makeplot)
else
    Wt = (Yt(2:end) - c0.*Yt(1:end-1))./(1-c0);
    [fitresult, ~] = fit( t(2:end), Wt, 'poly1');
    [temp] = coeffvalues(fitresult);
    b0 = temp(1); %a0=temp(2);

    Dc= 1;
    Db= 1;

    while Dc>0.01 || Db>0.01

        % STEP 2
        Yt1 = Yt(2:end) - b0.*t1;
        c1 = autocorr(Yt1,1); c1 = c1(2);
        if c1<0.05,
            c = c0;
            b = b0;
            break
        else
            Wt = (Yt(2:end) - c1.*Yt(1:end-1))./(1-c1);
            [fitresult, ~] = fit( t1, Wt, 'poly1');
            [temp] = coeffvalues(fitresult);
            b1 = temp(1); % a1=temp(2);

            Dc = abs(c1-c0);
            Db = abs(b1-b0);

            disp(Dc)
            disp(Db)

            if Dc<=0.01 && Db<=0.01
                c = c1;
                b = b1;
            else
                c0 = c1;
                b0 = b1;
            end
        end
    end
end

close all
Wt = (Yt(2:end) - c.*Yt(1:end-1))./(1-c);
[taub tau h sig Z S sigma sen n senplot Cllower Clupper D Dall C3] = ktaub([t1 Wt(:)], alpha, wantplot)
end
```

end

For function ktaub, see:

Burkey, Jeff. May 2006. *A non-parametric monotonic trend test computing Mann-Kendall Tau, Tau-b, and Sen's Slope written in Mathworks-MATLAB implemented using matrix rotations.* King County, Department of Natural Resources and Parks, Science and Technical Services section.

<http://www.mathworks.com/matlabcentral/fileexchange/authors/23983>

6. L85: Please expand “non-autocorrelated residuals” to “non-autocorrelated residuals (see Methods)”.

Authors: Corrected throughout the manuscript.

7. Fig. 1A: What does the GWP trend pattern look like? Is an increasing GWP seen everywhere over the global oceans? If not, I think such trend pattern is worth presenting, at least in the Supplementary Material, and regional mean WP time series should be shown for each region of the same trend direction.

Authors:

This is a critical point. We agree with the reviewer. This information was necessary to provide a better interpretation of the results (also in line with comments by Reviewer 3).

To address this point we have included new figures and added new analysis to the article. However, these new figures have also required additional reorganization of the text. These additions are described below:

First, we have included a Figure showing the regional trends by ocean basins. Figure 1 now shows the time series of WP and the 10-year moving average for each ocean basin. We have also calculated equivalent time series by latitudinal bands and included it in the Supplementary Information.

Figure 1. Spatial mean annual WP calculated globally and by ocean basin. The dashed lines represent the 10-year moving averages. The Southern Ocean is defined between latitudes of 40°S and 80°S. The mean regional WP is calculated as the spatial average of each historical wave power time series (see Methods). The time series calculated by latitudinal bands can be found in Supplementary Figure 1.

Second, we have included the spatial trends in a new Figure in the main manuscript. Figure 6 (below) now represents the spatial trends in WP. We opted to express the pattern in % change because is more illustrative of the changes than the absolute magnitude in Wave Power (in kw/m), and also because this percentage change allows a direct comparison with the other studies, in particular (Young et al. 2011).

Figure 6. Spatial trend (percent change per year) in mean WP from 1985 to 2008. Hatched areas represent points that are statistically significant at the 95% confidence level according to the Mann-Kendall test and the Wang and Swail method for autocorrelation (see Methods). The trends are calculated for the period 1985-2008 (period with satellite-derived wave data) for comparison with 38. Supplementary Figure 4 shows the spatial trends for other periods in the historical record.

We limit the period in Figure 6 to 1985-2008 because we wanted to allow a direct comparison with the trends found in (Young et al. 2011) for Hs and wind speed from satellite observations. This comparison is critical to explain the changes we find and the comparability between the results. We find similar patterns as for the mean and 90th percentile trends for the significant wave height (Hs). **1% in Hs trends in Young et al represents over 2% in WP, in line with the patterns identified in Figure 6.** This comparison is now explained in the Discussion section, Lines 308-322.

We have also calculated trends in different temporal spans: by 20-year blocks, by reanalysis eras and for the full period. For extent restrictions, these results have been included in the Supplementary material. In particular, Supplementary Figure 4 (below) represents this spatial pattern in absolute (kw/m) and relative terms (% change) for the whole period of the reanalysis and for the satellite era (1979-2008).

Supplementary Figure 4. Mean change in WP over the historical time span (1948-2008) expressed in absolute values per decade (a), percentage per year (b), and percentage per year but only for the time span covering satellite-derived wave measurements (in other words, after 1985). The hatched areas represent values that are not significant at the 95% level.

8. L106: Correlation is between two time series. Or are you talking about autocorrelation? It is inappropriate to say “The correlation of each time series is also significant...” Maybe ““The correlation of each pair of the time series is also significant...”?”

Authors:

The text has been rewritten as:

Line 175 - *GWP and SST are also correlated globally on a seasonal scale (Figure 3).*

9. L108-109: I would modified the sentence to “... analysis shows that the maximum correlation is 0.76 and occurs between SST in a season and the GWP one season after (time lag =1).”

Authors:

Corrected. Now Lines 177-178:

‘ A lagged-correlation analysis shows that the maximum correlation is 0.76, between an SST during a given season and the GWP one season after (time lag=1; Supplementary Fig. 3). ‘

10. L137-138 and L142-144: Kumar et al. (2016) also studied global impacts of natural climate variability such as ENSO, NAO, and PDO on extreme significant wave height, which is worth citation here. Wang and Swail (2001) also related NAO to wave extremes.

Authors:

Agree, we have added these key references.

11. Fig. 3: What does the shading scheme represent, correlations?

Authors:

The shading was representing major El-Niño events. However, this representation has been changed in the revised version, following suggestions from the Editor and Reviewer 1. Now we annotate in the Figure the events of El-Nino by their years of occurrence.

12. L169: Please add a reference for the statement “Outside their generation zone, swells are not significantly affected by surface winds.”

Authors:

Corrected. (Alves 2006) has been added to the text. The explanation on this regard has also been improved. Lines 247-248.

13. L214-215: Many more studies on projected future changes in wave climate can be cited here. At least some of the studies listed in the References below should be cited here.

Authors:

We have added the suggested references.

14. L215-216: Will all SST variations (both increase and decrease) lead to increased wave height? I think you meant to say “SST changes in the ... will lead to wave height changes over ...”.

Authors:

We have rewritten the text to better indicate how the pattern in the Figure is consistent with recent research on climate projections. We also specify ‘warming’ as this is consistent with the references we provide, for which we have also added new relevant references.

(Fan et al. 2013) use a coupled system to investigate the mean and extreme (99th percentile) surface wind and wave climate changes associated with SST–sea ice anomalies with various models for the end of the twenty-first century. The SST anomalies in the eastern and central Pacific exhibit a more

pronounced warming with patterns that resemble El Niño SST anomalies and with similar wave fields as in El Niño situations (Fan et al. 2013).

(Mentaschi et al. 2017) show that ENSO shows a significant tendency to increase (Figure 4k) expressed by an intensification of the ENSO pattern and a shift of climate toward El Niño conditions (Cai et al. 2014).

Given the pattern we obtained in the new Figure 8, and correlations with El Niño index in the new Figure 7, now the text reads as - Lines 386-393:

‘Climate projections for the end of the century show that SST warming in the tropical Pacific will lead to increases in wave height and extreme wave energy levels at mid- to high latitudes over the Southern Ocean and central north Pacific (Fan et al. 2013; Wang et al. 2014; Shimura et al. 2015; Mentaschi et al. 2017). These changes are triggered by the intensification of the SAM, ENSO and the NAO (Mentaschi et al. 2017), with a poleward shift in storm tracks and an enhanced westerly jet (Wang & Swail 2006; Fan et al. 2013; Hemer et al. 2013; Wang et al. 2014, 2015). The historical patterns we find in Figure 8-a help explain these effects of SSTs increase on the wave climate, where similar SST warming in the tropical Pacific and mid and north Atlantic led to increased WP in the same basins (Southern Ocean and the north Pacific) in the historical period (Figures 7 and 8). ‘

15. L217-219: Did you analyze this? or Ref. 48 did this? Please clarify.

Authors:

We have rewritten the text to be more specific and avoid misleading interpretations (see previous comment).

16. Table 1 and elsewhere: I think it is less confusing to use lagged-correlation in place of cross-correlation, because the latter is usually used for the correlation between two time series.

Authors:

Agree. It has been corrected.

17. L419: “to remove”? Maybe “to set aside”?

Authors:

The text has been rewritten as:

“Hurricanes and typhoons in the satellite data are identified applying an outlier filter based on weighted least squares at a 95% significance levels and not considered in the calibration (Minguez et al. 2011).”

18. L435: “...in each signal” \diamond “...in each data time series”.

Authors: Corrected.

19. Table 2, column header: “Autoregressive terms” \diamond “Number of autoregressive terms”. “Moving Average terms” \diamond “Number of moving average terms”.

Authors: Corrected

20. L511: Change to “...randomness in a variable. The entropy, $H(X)$, for the variable X of ...”

Authors: Corrected

21. L516: “since $H(X)$ is only equal to $H(X,Y)$ ” should be “since $H(X,Y) = 0$ in this case.”

Authors:

To provide a clearer description, the sentence has been corrected and rewritten as:

‘ If X and Y are independent, the total entropy of the system, $H(X, Y)$, would be equal to $H(X) + H(Y)$, in which case the MI value would be zero, indicating that X does not contain information about Y , and vice versa. ‘

22. L522: “Figure 5C and d” \diamond “Figures 5C and 5D”.

Authors:

Corrected. We also note that the formatting and numbering of the Figures have been modified to meet the Journal’s format requirements.

Supplementary Material:

General comment from Authors:

The Supplementary Material has been extensively revised. New Figures and other materials have been added to complement the discussion in the main manuscript, and to provide deeper analysis and interpretation of the results.

L11-12: Can cite Wang et al. 2013 and Wang et al. 2006 here (about inhomogeneities in several reanalysis datasets).

Authors:

References added.

L14: Can cite Wang et al. 2006 here (about inhomogeneity of NCEP reanalysis)

Authors:

Reference added.

L90: Kumar et al. (2016) also studied global impacts of natural climate variability such as ENSO, NAO, and PDO on extreme significant wave height, which is worth citation here. Wang and Swail (2001) also related NAO to wave extremes.

Authors:

References have been added to the Supplementary but also to the main paper. Given the lack of restrictions in the number of references in the supplementary, we have also added Wang et al. (2011) and others.

Suggested References:

- Caires, S., V. R. Swail, and X. L. Wang, 2006: Projection and analysis of extreme wave climate. *J. Climate*, 19 (No. 21), 5581–5605.
- Kumar, P., S.-K. Min, E. Weller, H. Lee, and X. L. Wang, 2016: Influence of Climate Variability on Extreme Ocean Surface Wave Heights Assessed From ERA-Interim and ERA-20C Reanalyses. *J. Clim.*, 29, 4031-4046.
- Wang, X. L., Y. Feng, V. R. Swail, and A. Cox, 2015: Historical trends in the Beaufort-Chukchi-Bering Seas surface winds and wave heights, 1970-2013. *J. Clim.*, 28, 7457-7569.
- Wang, X. L., Y. Feng, and V. R. Swail, 2015: Climate change signal and uncertainty in CMIP5-based projections of global ocean surface wave heights. *JGR-Oceans*, 120, 3859–3871.
- Wang, X. L., Y. Feng, and V. R. Swail, 2014: Changes in global ocean wave heights as projected using multi-model CMIP5 simulations, *Geophys. Res. Lett.*, 41, 1026-1034.

- Wang, X.L., Y. Feng, G. P. Compo, V. R. Swail, F. W. Zwiers, R. J. Allan, and P. D. Sardesmuks, 2013: Trends and low frequency variability of extra-tropical cyclone activity in the ensemble of Twentieth Century Reanalysis. *Clim. Dyn.*, 40, 2775-2800.
- Wang, X.L., Y. Feng, and V. R. Swail, 2012: North Atlantic wave height trends as reconstructed from the twentieth century reanalysis. *Geophys. Res. Lett.*, 39, L 18705.
- Wang, X. L., V. R. Swail, F. W. Zwiers, X. Zhang, and Y. Feng, 2008: Detection of External Influence on Trends of Atmospheric Storminess and Ocean Wave Heights. *Climate Dynamics*, 32, 189-203.
- Wang X. L. and V. R. Swail, 2006: Climate change signal and uncertainty in projections of ocean wave heights. *Climate Dynamics*, 26, 109-126.
- Wang, X. L., V. R. Swail and F. W. Zwiers, 2006: Climatology and changes of extra-tropical cyclone activity: Comparison of ERA-40 with NCEP/NCAR Reanalysis for 1958-2001. *J. Climate*, 19, 3145–3166.
- Wang, X. L. and V. R. Swail, 2002: Trends of Atlantic wave extremes as simulated in a 40-year wave hindcast using kinematically reanalyzed wind fields. *J. Clim.*, 15, 1020-1035.
- Wang, X. L. and V. R. Swail, 2001: Changes of Extreme Wave Heights in Northern Hemisphere Oceans and Related Atmospheric Circulation Regimes. *J. Clim.*, 14, 2204-2221.

Authors:

We appreciate the suggestions regarding the references. They have been added throughout the manuscript, methods and supplementary materials, leading to a general improvement of the article.

Reviewer 2

(Remarks to the Author):

I have just finished reviewing the manuscript 'Uncovering a new global signal of climate change: evidences of a stronger global wave climate associated to ocean warming'. The manuscript deals with the interplay between sea surface temperature and wave power, a very interesting topic per se. The research gains additional interest given (i) the (large) spatial and (long) temporal scales; and (ii) the fact that it resolves teleconnections among ocean basins. Finally, the paper is scientifically robust and based on sound datasets and methodologies and interesting discussion of the output.

Authors:

We appreciate the comments and suggestions provided by the reviewer.

Given all the above I read the manuscript with pleasure, however I have some issues about the presentation, therefore I cannot recommend for direct publication.

My main issue is that the paper does not appear to be prepared for Nature Communications and I find that the authors could improve a lot the manuscript to that direction. For the moment the manuscript is incomprehensible without the methods and the SI and the reader needs to do several iterations along of each of them to gain insight on the work. I think this could be improved substantially.

Authors:

We acknowledge these concerns. The article has been extensively revised and reorganized to address them. The Figures have been edited and improved for clarity too. More results and discussion have added to provide more robust results and interpretation. We refer to the general comments at the beginning of this document, but a summary follows:

We agree with the reviewer the previous version lacked specific methodological descriptions that made difficult to understand the results. This has been corrected. We have extensively reorganized the content, with special attention to the Methods. We have also included many parts of the methods in the main body of the manuscript (following the journal's suggestions) but also critical information that was previously in the Supplementary information has been added to the Methods and throughout the text. We include now brief descriptions of the relevant methods accompanying the Results section, to facilitate the understanding of the manuscript without the methods.

The manuscript language and overall organization have also been modified to improve the overall flow and readability. We have restructure the article following the structure suggested by the Editor. Now, the last paragraph in the Introduction describes briefly the methods and provides a brief summary of the major results and conclusions. This follows the journal's indications and improves the comprehensibility of the work.

The methods and the Supplementary Material have also been extensively edited. Specifically, much of the information previously contained in the Supplementary Material has been included in the main text (for example, new Figure 5, following the suggestion of the reviewer).

The authors could dedicate a paragraph after the introduction providing an overview of the methodology.

Authors:

The Introduction now includes a brief overview of the methodology and results (L180-195), following also the Editor's suggestion.

After the introduction we have also added new text (Lines 89-104) that describes the methods before presenting the results. A summary of methods include:

1. First two paragraphs in the Results, Lines 107-134, now provide crucial information on the different wave data used and how the time series of wave power are calculated, before we can introduce Figures 1 and 2.
2. We specifically refer now to the correlations between what time series and in which periods.
3. A description of mutual information analysis is now preceding the Figure and the presentation of the results.
4. We also explain the rationale for the sequence of steps and why we analyze first global changes and then move to regional ones, to explain them through teleconnections (using climate indices and ocean sub-basin time series).

Phrases like 'as applied in 12' found in line 117 could be the way to go for other journals, but my opinion is that there are more appropriate ways to present the work in the present journal; which would allow the reader to get the main message without having to read several other papers.

Authors:

We agree. The revised version includes relevant descriptions throughout the text to better explain the steps we take before presenting the results.

In the specific example pointed by the Reviewer, we provide a better context on the approach and why we use it to study the statistical dependency between SST and the GWP. A basic description of the approach is included before presenting the result of the test, with the methods providing a more thorough description. Now this paragraph reads (Lines 183--190):

"...This technique is applied because it has been shown useful to determine the influence of SST on hurricane intensity based on analysis of time series in an equivalent application³⁵. In information theory, the mutual information (MI) of two variables is quantified to represent the measure of independence of the two variables³⁶. In the context of the present work, MI can be thought of as a measure of SST

information that is shared by GWP and is calculated from the entropy in the time series, a measure of the randomness of the variable (see Methods). To quantify MI, we estimate.”

We also note that the Methods have been edited and completed with relevant information that was originally in the Supplementary Material.

Also references are in general used like in the [author, year] format and I think for Nature style (number) they should be incorporated in a different fashion in the text. I am also not sure but I have the feeling that Nature communications has a limit on the max number of references.

Authors:

The references have been edited to comply with the journal style.

Regarding the number of references, the journal guidelines recommend a total of 70. This number of references is slightly exceeded in the revised version. However, we consider many of these references are necessary (particularly for the deeper interpretation of results and methods). We have included new ones compared to the original manuscript but deleted others from the previous version (see also comments and references suggestions made by Reviewer 1).

We now compare our results with (1) historical changes in other wave parameters and (2) explain future changes based on projections through the inter-regional correlations. This has required some new supporting references. However, we have tried to limit the number of references and select the most relevant.

When the authors discuss the results they could include only the relevant information. For example, already from line 48 it becomes confusing for the reader to follow the results with the trends from the different datasets. In my opinion discussing the trends from different datasets should not be the point, but the authors should focus on the most reliable information. The same applies for the SST datasets.

Authors:

We have modified the Results to focus on the most relevant information. Now, we describe the results for the longest time span, i.e. 1948-2008, because it is the main contribution of this article. However, we opted to maintain the results for the satellite era in parenthesis, i.e. 1992-2008, because they are relevant for interpretation of the analysis and to some discussions about teleconnections (see following comments).

However, the number of references to different datasets and periods have been revised and reduced.

We also think that the new text and re-organization of the manuscript help address this point too. The new structure of the paper with a substantially revised Introduction (e.g. new narrative, presentation of the research question and hypothesis, concluding with a paragraph that summarizes the methodology and main results), and two independent Results and Discussion sections facilitates the readability.

For example, the specific paragraph explaining the wave datasets also clarifies the differences between them and how each time series is used (Lines 118-134). We use the satellite-derived data and the additional wave hindcast data to validate the signal and variability in the long-term dataset. We also point to the need of long-term time series to identify sustained changes associated to climate change.

The presentation of the MI results was another aspect that forced me to go through the methods and the manuscript several times. First of all, it is not clear why the authors selected this method to test dependence among the two variables, since there are other more common and simpler to interpret statistical methods. The authors could elaborate a bit on that in their reply.

Authors:

We use information theory because it is a technique that has been used successfully to explain the dependency of Global Hurricane intensity with SST, in a very similar context, by Hoyos et al (2006). We think this technique was particularly relevant to relate SST with GWP.

However, the rationale for the technique and how it is used has been clarified in the revised version of the text (Lines 183-197 in the Results).

However, we note that we use this technique to complement other methods, such as: linear correlations and regression of time series, correlations of the non-autocorrelated residuals, and trend identification (see also comments from Reviewer 1 in this regard).

Figure 3 is aesthetically good, and tries to provide some basic concepts, but is a bit confusing. In the upper row there are patches for values beyond -1, 1 but it is not explained why. The lower map plots have color variations which are not explained by any colorbar. Moreover, it is not clear whether the spatial extent of the colored areas refers to a specific year, or some mean behavior. Moreover, the authors mention spatial influence of the indices to WP, but no evidence is shown. So in this format it doesn't really help a lot.

Authors:

We agree, and the Figure has been substantially edited. The original Figure was intended to signify a schematic representation of the ocean regions with correlation of WP with each climate index. The revised version provides more clear and quantitative information, also in a more consistent format, which is particularly relevant given the related discussion of the teleconnections found in this paper.

We have modified the Figure and reorganized the panels. The new Figure (now Figure 7) contains 4 panels, two of them representing the time series of each standardized climate index. The blue and red shadowed areas represent events that are considered 'moderate' (values above 1.0 of standardized value).

The spatial patterns correspond to areas correlated with the climate indices time series. This is now better explained in the text and the Figure caption.

Figure 4 together with Table 1 summarizes the main findings of the work and give a good insight in the teleconnection patterns. There are differences between the two datasets and the authors provide some explanations, but still some aspects have not been clarified. Could the fewer teleconnections in the satellite era be due to the shorter dataset?

Authors:

We agree that further explanations were needed to accompany and support these main findings.

We have added new results and performed further analyses. We calculated the trends and decadal changes in blocks of 10 and 20 years to better explain these changes. We have included two Figures in the Supplementary Information (Supplementary Figures 5 and 6) that represent the interdecadal changes in SST and WP. We also discuss these decadal changes in relation to the spatial trends and inter-regional correlations in the text. We considered this information was very relevant to be included in the main text, but we opted to maintain the Figures in the supplementary material given the restrictions in number of visuals.

We find two main reasons:

1. The shorter time series of the satellite era are less able to capture long-term effects and are more affected by the interannual variability.

2. The reinforcing of the Atlantic and weakening of the Pacific teleconnections are explained by the interannual variability during the satellite era. This effect is portrayed in the decadal changes in WP (Supplementary Figure 5) and SST (Supplementary Figure 6). There is a strong interannual variation during the second decade of the satellite era (1989-1998). El-Niño in the Pacific and the positive Atlantic Multidecadal Oscillation in the Atlantic, drive most of the correlation observed in the Pacific and the Atlantic with other ocean basins. These historical teleconnections patterns and the decadal changes identified in the new Supplementary Figures are consistent with the behavior identified in Figure 7 for El-Niño and the Atlantic Multidecadal Oscillation climate indices.

This explanation is very relevant for the interpretation of the results, but was not adequately attended before. We have included these explanations in the revised text. The fewer teleconnections we find for the satellite area are now explained in a specific paragraph in the Discussion - Lines 375-383:

‘ The smaller number of teleconnections in the satellite era (Figure 8-b) relative to the entire dataset (Figure 8-a) are likely due to a shorter record and the presence of strong interannual variability during those decades, which are dominated by the 1998 El Niño in the central Pacific and the AMO in the Atlantic. Some teleconnections are also likely hidden in the satellite era in the southern hemisphere due to decadal changes associated with the SAM (the principal mode of variability for extratropical atmospheric circulation in the southern hemisphere), which enhances westerlies across the Southern Ocean during its positive phase and increases and rotates wave energy clockwise as the storm belt intensifies ¹¹. The strong interdecadal variations in WP and SST also reflect the difficulty to find teleconnections and trends over short periods, such as the satellite era, and highlight the need to use long-term records, as in Figure 8-a. ‘

It is also worth noting that this point speaks of the need of long-term series for the identification of trends and significant consistent patterns. When we calculated the trends by periods of 20 years or for the satellite era, the trends were masked by the interannual variability, particularly during the decades mentioned above. Nevertheless, the patterns we identify for the satellite era are consistent with the changes identified in Young et al (2011) for high percentiles of significant wave heights derived from satellite observations (see responses to Reviewer 1, revised text and new Figure 4).

The next important point the authors make is the correlation between SST and WP. First of all this I find confusing that two equations are used for the correlation. I also believe that Figure 1 of SI should be included in the manuscript.

Authors:

The Figure has been edited and added to the main manuscript.

At the risk of adding confusion with two equations, we have maintained both because each one captures different effects in the time series (both represented in the new Figure 5, but represented at the end of this comment in an auxiliary Figure). The regression on the long-term time series gives the average expected change in the GWP with SST warming (Figure 5-a). However, by regressing the rates of change in each time series (i.e. annual variations), we obtain the expected change in GWP in a given year, given a certain change in the SST anomaly. This second regression represents the variations in GWP associated with the variability in the temperature. This has been explained in the text. These distinctions have been clarified in the text, which now reads:

Lines 201-206: *‘ ... We previously showed that the not only the long-term time series of the GWP and SST but also the non-autocorrelated residuals (a measure of the variability in each signal) were also highly correlated (Figure 2-c). We estimate that GWP can change annually by 1.2 10⁴ kilowatts m⁻¹, on average, for an annual change of 0.1°C in SST. This represents a ~4.3% annual change in GWP (1.2 10⁴*

kilowatts m⁻¹ over the mean GWP during 1948-2008), equivalent to a global long-term change over a decade (i.e. 4%, Figure 1).'

Figure 5. Regression between the SST anomaly and GWP. (a) Regression between the global time series of SST anomalies (°C) and GWP (kw/m); (b) regression between the annual rates of change in SST (°C/yr) and GWP (kw/m/yr). The solid red lines represent the linear regression lines (equations are noted in each plot). The red dashed lines represent the 95% confidence intervals for each regression line. Both regression lines are statistically significant at the 95% level. The blue dots represent the data for the period 1948-2008.

Auxiliary Figure. Time series of each variable used for the two regression models. The two first panels show the time series of the WP and SST, while the lower two panels represent the annual rate of change ($X_{i+1} - X_i$).

Apparently, such a regression fit would have severe implications under global warming, however such an increase has not been found by studies dedicated to wave projections. Could be a limitation of GCMs, or other factors come to play and impede the direct effect of SST on WP? If that would be the case shouldn't there be an increasing trend in WP at least for the shorter future? Discussing more in detail the possible reasons for that discrepancy between present findings and future trends would be very important, as this is a key point.

Authors:

This is a relevant point also raised by Reviewer 3. We have added new text in the Discussion that address this critical point. Our teleconnection patterns are consistent with (1) historical trends for other wave parameters (for example, 1% increases in Young et al, 2011, are comparable with our 2% results for the WP), and (2) projected patterns of change in the wave climate.

Only the study by (Mentaschi et al. 2017) study global trends in extreme wave energy flux in the 21st century for the future (which is comparable to WP), but only along coastlines. It will be expected that the average wave energy will be larger in the open Ocean than estimates of 30% calculated for the continental coastlines. Swell energy analyses show that the Southern Ocean swells dominate throughout the global Ocean (Fan et al. 2014). If averaged for the global Southern Ocean, the increases should be larger than closer to the coastline (e.g. see Patterns of WP in Reguero et al 2015).

Most of the remaining studies have focused on significant wave heights and mean and high percentile values. This makes a direct comparison difficult. However, our spatial patterns are consistent with results in (Hemer et al., 2013), and (Fan et al. 2013) for multi-model ensembles. Storms contribute proportionally

more to the annual WP. Projections in Hemer et al (2013) for H_s are in the order of 10% but for the mean conditions. However, there are also scientific consensus on larger changes in the high percentiles of H_s , which contribute proportionally more to WP. Mori et al., (2010) also show 6%–9% increases in mean H_s in the Southern Ocean. This indicates that the U10 and H_s increase in the Southern Ocean are robust across different studies (method, model, or scenario) regardless of the magnitude of variations among them.

Furthermore, (Fan et al. 2013) find the most robust response observed is the increase of U10 and H_s in the Southern Ocean across seasons due to the poleward shift of the jet streams and suggest that larger percentiles of H_s will increase more than double than mean conditions. Similar changes were also observed by Wang and Swail (2006) through their statistical projections but with smaller magnitude.

Our range of estimates, for the different Representative Concentration Pathways, is 32-122% increase by the end of the century (regression in Figure 5-a). (Hemer et al., 2013) finds that mean wave conditions will change in the southern high latitudes by 10% and (Fan et al., 2013) points to twice as much increase in the larger waves as compared with the mean conditions. A 20% increase in the waves of storm conditions represents a 44% increase in GWP, in our range of values. Furthermore, Hemer et al uses ensemble models, and Fan et al the A1B IPCC AR4 scenario, which predicts lower warming than the RCP 8.5.

Assuming the historical regression -which has its limitations and we acknowledge these caveats in the text-, and the projected values of 0.6°C (RCP2.6) to 2.0°C (RCP8.5) for SST by the end of the century, this would represent between 36% to 120% increase in the GWP with respect to 1986–2005. An equivalent change in H_s would be 16% to 48% in H_s (leaving the wave periods unchanged). The lower range is comparable to projected changes from ensembles, but the RCP8.5 estimate seems overestimated.

This point is now addressed in the Discussion – Lines 299-322:

‘...Based on a linear regression between the global time series of GWP and SST (Figure 5-a), a 0.06-0.1°C increase per decade represents a GWP increase between 4.2 and 6.9% per decade (calculated with respect to the mean WP during 1948-2008). Projections of warming in the top 100 meters of the ocean range from approximately 0.6°C (RCP2.6) to 2.0°C (RCP8.5) by the end of the century, relative to 1986–2005 (IPCC 2013). For these projected increases in SST, the regression of the time series gives increases in GWP between 32 to 122% (relative to GWP during 1986–2005; an equivalent change in wave heights would be 16% to 48%, leaving other factors unchanged). These estimates can be compared with projected future changes in the wave climate.

The average multimodel projected changes in significant wave heights (H_s) are over 7.1% of the global ocean, but occurring predominantly in the Southern Ocean, with increases of up to 10% in (Hemer et al. 2013), which is the most energetic basin. This would lead to an approximate increase in WP of 21% when considering only the mean value of H_s (more pronounced because it relates to the square of H_s). However, projections indeed suggest that the high percentiles of H_s , which contribute proportionally more to the WP (with the square of H_s), will vary by up to twice as much as the changes in the mean values (Fan et al. 2013); a behavior consistent with the historical satellite-derived trends found by Young et al (Young et al. 2011). This will represent a 44% increase in GWP, in the range of our estimates based on regression with SST (32-122% relative to 1986–2005, depending on the RCP). We also note that these estimates have been provided for ensemble projections and lower warming scenarios than the RCP8.5 for which we find the most extreme increases (122% for a 2°C SST warming). Ensemble projections of wave energy flux for the end of the century show also a significant increase (up to 30% at a 100-year return level) for the majority of coastal areas in the southern temperate zone (Mentaschi et al. 2017). However, these values only correspond to coastal areas and not the whole basin, and it should be expected that the increase in offshore values in the open Ocean will be larger than in the coastal areas

because wave energy is lower closer to the continents(Reguero et al. 2015), also in agreement with our estimates. ‘

We also caution about a direct extrapolation, particularly for long horizons of the century, in Lines 324-335:

‘Although the projections of GWP based on direct regression with SST are in the order of magnitude of dynamic projections of wave climate, direct regression with SST needs to be taken with caution. Semi-empirical relationships have been used before to explain the responses in variables related to global warming, for example, sea level rise (Vermeer & Rahmstorf 2009). However, there are important factors that prevent the direct extrapolation between SST and GWP at the end of the century. First, there are underlying uncertainties in the climatic mechanisms and dynamic effects of climate change that are not reflected in the historical time span. These uncertainties include the effects on wind, and thus WP, due to the weakening of the Atlantic Meridional Overturning Circulation (11% in RCP2.6 and 34% in RCP8.5, on average) and the changes in Arctic sea-ice cover, which can modify where waves are generated and how they reach coastal zones. Furthermore, we have also shown in Figures 7 and 8 that variations in interannual climate patterns are a key factor in WP variability, and these patterns could change in the future in terms of intensity and frequency (Mentaschi et al. 2017). ‘

A less important point... It is confusing that in the several tables discussing correlations between SST and WP the word autocorr appears. Is this correct, since we are referring to two variables?

Authors:

The use of this term is incorrect. We have corrected it in the main text and in the Supplementary Tables, and clarified that the regression is established between the two time series. This clarification was important, particularly given the new method to avoid ‘autocorrelation’ in the time series when assessing the significance of the trends (see comments of Reviewer 1 and new implemented method).

We also note that we have simplified the information provided in Table 1 (reduced by ~50% the number of values provided originally).

Reviewer 3

(Remarks to the Author):

Review of the NCOMMS manuscript “Uncovering a new global signal of climate change: evidences of a stronger global wave climate associated to ocean warming” by Reguero et al.

Recommendation: Reject

Fundaments:

The paper presents a study where the supposed increase in global wave energy flux is related to the recent increase in SST. The study is based in the GOW wave data set and on a NOAA global SST composite data set.

Relating SST to wave fields, directly, has a certain degree of novelty (some other authors have done it for other parameters, through). Nevertheless the manuscript has some flaws, lacking depth, that prevent it, in my opinion, to be accepted in such a high level journal.

Authors:

We appreciate the comments and points made by the reviewer. We have addressed the flaws in the original submission and provided new results and in deep analysis. New Figures and other materials complement the significantly revised Discussion in the main manuscript, which overall provides more deep analysis and interpretation of the results, as requested by the Reviewer. All these modifications have led to an improved version of the manuscript and a better explanation and interpretation of the results. We summary the main changes in the following (see also summary at the beginning of this document).

We have focused on supporting our argumentation through a clearer and more logical narrative from the onset. We also appreciate recognizing the novelty of studying and relating SST to wave climate changes. It has been highlighted in the Abstract, Introduction and Discussion in a clearer way in the revised version. We now point to the contributions, which the reviewer also acknowledges, from the onset of the article and also avoid misleading interpretations. This includes, among other major changes, the change in title to one that is less interpretative but one that better summarizes the findings. The new Figures and results also provide further insight and depth in the results (also complemented by new analyses included in the supplementary material, see also comments by Reviewer 1 on this regard).

We have also made the statements more quantitative; several statements were done qualitatively and using citations without comparisons. The new narrative and structure of the Introduction and Discussion, an intensive language and Figures editing, substantial consolidation of the methods and supplementary materials (attending also comments from Reviewer 2), new results and discussion points, and other significant changes in the Results and Discussion sections, all help now to present more clearly the rationale and the contribution of the article.

In particular, the flow of ideas to sustain the results has been extensively revised. For this, we have added new important results and rewritten the Discussion to explain how the global changes in SST and WP are explained through regional patterns and are consistent with other historical and future changes in waves and winds (see previous comments).

The introduction and Results also present our rationale to support the global changes we find and how SST and WP are connected, globally and regionally. While historical changes have been found for wave parameters (mostly different statistics of the significant wave heights), we propose to use the wave power because it better reflects the energy transferred to the ocean motion, as a consequence of the atmospheric circulation. We explain this now in a paragraph before presenting the Results (for extent limitations, we could not include this explanation in the 1,000-words Introduction). We then describe the methods and the results found for the long-term changes in WP and SST.

However, this long-term trend occurs with teleconnections between SST and WP for certain ocean sub-regions. These connections have been established before based on correlations of climate indices with several wave climate parameters. We extend other studies based on climate indices to WP in Figure 7. This Figure has been extensively edited for clarity and consistency (see also comments from Reviewer 2).

The correlation with climate indices is an indication of regional relationships between SST and WP. However, they are region specific. For this reason, we also study inter-regional correlations by ocean sub-basins. The teleconnections between ocean sub-basins are key to explain the long-term trend in the global wave power. We explain and dwell further on how these connections are consistent with previous climatic research (including known effects on wave climate established for certain basins and other wave parameters), but also predicted future changes. That the connections we find are in agreement with this body of knowledge for the different ocean basins is a further proof of a global relationship between historical SST and WP.

To conclude, we also note that the Supplementary Information has been extensively revised.

The text is not uniform (clearly it has been written by several hands).

Authors:

The text has undergone extensively revision and edits (including by the scientific and language editing service). We have also reorganized the sections and text, to improve the clarity and the flow of ideas.

Comments:

L22: variables or parameters?

Authors

Note that the Introduction has been extensively edited and changed.

In general, we refer to GWP and SST as variables through the manuscript when referring to the time series. We use 'parameters' when referring to wave climate statistics, like the significant wave heights. We have doubts if this is the most exact use in this context. We use the term '*variables*' because it represents an attribute or quantity, whose values vary depending on certain conditions. A parameter is very similar to a variable in that the value also varies but is normally describing a certain property of the variable (e.g. usually a parameter in an equation describing a model).

L22-23: what do you mean with "One for the global wave climate has not been found"? Several studies have looked at the impact of present climate change and projected changes in climate in the global wave field (on T_m , H_s , and some even on global wave energy flux), hence how can you state this?

Authors:

We agree that these changes have been found spatially for different wave parameters, and from different data sources. We better acknowledge this point in the new Introduction and refer to many of these studies throughout the manuscript (see also comments from Reviewer 1). This has been clarified in the new writing for the introduction (L41-53)

'Most analyses of the wave climate (i.e. the description of wave characteristics over a period of time) have focused on identifying historical trends in mean and extreme values of wave parameters, such as wave heights. Wave heights have on average increased in recent decades, particularly at the high latitudes of both hemispheres and for the extreme values as compared to the mean conditions (Wang & Swail 2001, 2002; Wang et al. 2009; Young et al. 2011). Satellite-based altimeter measurements from 1985 to 2008 reveal increases of 0.25% per year for the 90th wave height percentile and 0.50% per year for the 99th percentile, in both hemispheres (Young et al. 2011). Climate reanalyses also show significant increases in extreme wave heights at the high latitudes of the southern hemisphere (Hemer et al. 2010), 0.25-0.9% per year for the 90th percentile in the north Atlantic and the north Pacific, and decreases in the midlatitudes (Wang & Swail 2001; Dodet et al. 2010). However, not only wave heights have been changing, wave periods have also increased (Sterl & Caires 2005), and direction of waves shifted, for example in the southern ocean (Hemer et al. 2010) and in the north Atlantic (Dodet et al. 2010). However, changes in the global wave energy have received less attention, particularly in the context of climate change (Reguero et al. 2015). Despite the changes detected in different wave parameters, a global and long-term time series of the effect of climate change in the global wave climate remains undetected.'

We also note that the spatial trends for the historical WP (also in the context of SST warming) are another novel aspect of this study (not previously included, see comments from Reviewer 1).

L25: could? Here and in several parts of the text you use “could”. It seems you don’t have confidence in your results. Why could and not can?

Authors:

These expressions have been modified throughout the manuscript as suggested. The overall language in the paper has also been improved for clarity.

L28-28: Are you sure that you are the first to present the global wave energy flux as an indicator of climate change? And is it an indicator or a consequence of climate change?

Authors:

The text has been rewritten to address these two points.

We believe this paper is the first one presenting a historical global time series of the increasing WP. Furthermore, we explain this signal in relation to SST warming and through teleconnections, in alignment with research that found these spatial patterns for other wave parameters based on several climate indices.

For example, other studies have studied the wave energy flux, e.g. (Hemer 2010) for the southern ocean. Similarly, (Mentaschi et al. 2017) study projections of the wave energy flux along the shorelines for the end of the century, although this is not the scope of the present paper. We also added a comparison with these studies in the revised Discussion.

To our knowledge, our study is the first one providing this global signal, global trends in WP, relating them with SST warming and establishing teleconnections patterns globally (not only based on climate indices but on the wave generation areas). We think these are novel aspects of this study, but we build on previous research on historical and future wave climate. However, that these findings are in agreement with previous research should not be taken as weakness but an indication of the mechanisms here analyzed can point to direct connection between more energy in the waves associated to the historical global warming.

Regarding the use of ‘indicator’, the term refers to a climate change metric. However, we agree is a consequence of climate change and we have rewritten the text accordingly, although we avoid now interpretative assumptions.

Line 103: *‘This the first time that WP has been identified as a potentially valuable climate change indicator.’*

L37: what are remote sensing of changes? This whole sentence (L36-39 is confusing).

Authors:

The former opening sentences were not really helpful; they were widely known and do not seem essential to the more points relating to waves that follow. The opening paragraph has been completely changed (also to address some of the points raised by the reviewer from the onset).

L39: why broad?

Authors:

We have used the term *‘far-reaching’* instead. Line 32: *‘Surface gravity waves generated by winds have far-reaching implications for coastal areas. Wind-waves are an important contributor to...’*

L40: likely? See comment above regarding L25.

Authors:

We agree that the references provided and cited throughout the text provide unambiguous evidence of influence of climate change in the wave climate. We have modified the hypothetical tone of this expression to state this point more clearly. This has also been pointed out by Reviewer 1.

L44: Satellites are just platforms. They measure nothing. Remote sensing measurements would be the correct statement. What about the wave heights in the last 9 years (from 2009 to 2017)? Why don't you include these last 9 years?

Authors:

Agree. We have used 'satellite-based altimeter measurements' instead (as in Young et al 2011)

We only use the satellite observations in the coinciding period with the GOW data because this data is used to provide a third set of global wave power, to validate the long-term time series and its variability. Furthermore, we use the significant wave heights from altimetry but the wave period data is interpolated from GOW (in space and time), which does not allow to obtain results beyond 2008.

The text in the methods and Results has been rewritten to better explain it. For example, Lines 118-134 now explain the different datasets:

'To investigate trends in WP over time and its relationship with global warming, we first calculate long-term time series of WP and SST. Changes in wave climate have been studied through four types of data: buoy measurements, observations from ships, satellite-based altimetry records and numerical modeling. This study combines satellite altimetry and model results (validated with buoy measurements) to determine global WP (GWP) because observations from buoys and satellite altimetry do not provide continuous data over space and time. GWP was calculated using hourly time series of significant wave heights and mean wave periods from the altimetry-corrected global ocean wave (GOW) reanalysis for the period 1948-2008 (Reguero et al. 2012) (see Methods). To validate the long-term GWP from GOW, GWP time series are also calculated from other two sources: an independent global high-resolution hindcast (RaA13) that covers the period 1992-2012 and uses improved parameterizations for the model and high-resolution wind forcing (Raschle & Ardhuin 2013); and satellite altimeter data from 1992-2008 to calculate a third time series of GWP but based on wave height measurements instead of wave modeling results (see Methods). To calculate mean WP time series, the hourly time series of WP are aggregated by seasons and years, and averaged spatially globally and by ocean basin (see Methods). The GWP time series from GOW is highly correlated with the RaA13 (0.69) and satellite (0.66) time series. The variability in the global time series, as determined by the respective non-autocorrelated residuals (see Methods), are also highly correlated at 0.89 and 0.88, respectively.'

L48-49: Confusing.

Authors:

We have modified the text. The introduction has been substantially revised and edited for clarity.

L56: What has been used to investigate surface winds?

Authors:

The text has been rewritten to address this point.

L59-60: How can you state that “This may be an indication of a direct correlation between ocean surface warming and wave climate changes, but such a global connection remains to be determined”, right having stated two references (14 and 15) that do that? I don't understand what is new here.

Authors:

We agree the text was misleading. The new Introduction helps to clarify this point and also outlines better the contribution and findings of this work. For example:

- The paragraph in Lines 41-43 provides a review of studies that have found changes in the historical wave climate based on different observations and data sources.
- The paragraph in Lines 55-67 introduces the influence of SST on ocean-atmosphere circulation, in particular wind patterns, and presents our rationale.
- Lines 71-73: *Despite these effects of SST on the global wind patterns, SST has been largely overlooked when looking for a global signal of climate change in the historical wave climate.*
- Line 89: *In contrast with previous studies, we focus on the Wave Power (WP)...*
- Line 103: *This the first time that WP has been identified as a potentially valuable climate change indicator.*
- Lines 107-116: *WP represents the temporal variations of energy transferred from the atmosphere to the ocean surface motion over cumulative periods of time. WP can be a better indicator of long-term behavior of the global wave conditions than other parameters, such as the wave heights, because ...*
- Paragraph in Lines 89-104 explains the focus of the article and outlines the methods and results.

L62: Again could? And how are changes in SST related to storm patterns? References?

Authors:

The new writing in the Introduction addresses this point (e.g. Lines 55-67). We also describe this rationale when discussing the results in the Discussion section, to explain the results based on teleconnections.

The Introduction has been extensively modified for clarity and following the journals specifications. Now, we also provide a brief summary of methods and main results in the last paragraph of the Introduction, which also helps to address this point. Furthermore, relevant references have also been included in the Results.

L72: What do you mean with “WP is cumulative” in comparison with other wave parameters?

Authors:

This is an important point and has been better explained in the revised text (see also comment from Reviewer 2):

Lines 108-116: *'... WP can better represent the long-term behavior of the global wave conditions than wave heights because, first, WP includes information on significant wave heights (the mean value of the largest third of the wave heights during typically 1 hour, H_s) but also periods (T) (see Methods). Being proportional to $H_s^2 T$, WP therefore aggregates information of both mean and extreme conditions, but increases proportionally more with the latter (i.e., high percentiles of wave heights and periods). WP also represents the accumulated wave energy over certain periods of time, for example months, seasons and years, unlike other wave climate parameters, such as wave heights, that must be averaged over the same periods of time.'*

Furthermore, in Figure 1 and Supplementary Figure 1 we find an important different behavior for the north Atlantic, which sustains this assumption:

Lines 152-155: ‘ *Although the Atlantic Ocean has some of the largest extreme wave heights on the planet (Wang & Swail 2001; Vinoth & Young 2010; Izaguirre et al. 2011) (see the significant wave heights with a return period of 100 years in Supplementary Figure 2), its annual WP is the lowest of all ocean basins.* ‘

L74: ship observation or visual observations from VOS?

Authors:

The term has been corrected to ‘*observations from ships*’ (Line 120)

L75: Reference 22 is wrong in this context.

Authors:

We appreciate pointing this out. The text has been corrected.

Reguero et al (2013) analyzed trends in time series of different wave parameters but only regionally for Latin America and the Caribbean (not globally). However, Hemer et al (2010), Ruggiero et al (2010) and Menendez et al (2008) are also regional studies. We have corrected the ‘global’ statement to ‘changes in the wave climate’.

In this specific text, Hemer (2010) was substituted with Hemer et al (2010) which represents a large scale study.

We also note that other additional references have been added throughout the manuscript, but had to leave out many others due to limitations in the total number of references (see also comments from Reviewer 1).

L85: Are 0.66 and 0.69 “high” correlations?

Authors:

In this context, these values can be considered high correlations for annual global time series, based on (1) the short period of the satellite era; (2) the strong variability in the wave climate; (3) the fact that each time series is calculated independently and aggregated globally; and (4) each dataset has different spatial resolutions. Similar correlations have been found for other climate datasets, and with teleconnections patterns.

L128: energy or speed?

Authors:

Although both can arguably be used as synonyms in the context of long-term changes, Wang et al (2014) specifically states ‘*These wave height increases are primarily driven by increased SLP gradients and hence increased surface wind energy*’.

We would opt to keep this terminology to convey the idea of an increased energy budget in the climatic system, although wind-waves are generated by wind speed. We would like to note, however, that in other parts of the paper we refer to historical changes in wind speeds.

L129-130: “Long term changes and interannual patterns are closely associated”. Isn’t this a truism? Are they the same in this context?

Authors:

The new writing helps draw this distinction. We use *trends* for referring to long-term changes, as opposed to *interannual patterns* that refer to changes associated to climatic teleconnections (e.g. ENSO). However, the use of these terms has been clarified because the relationship between teleconnections and the long-term trend are an important aspect of this work. We also limit the use of '*interannual variability*' and use '*teleconnections*' when referring to changes that occur between years and between ocean sub-basins.

We also note that new supplementary analysis provides more insight into these decadal changes and its relationship with long-term changes and detection of trends for a few decades (see also comments to Reviewers 2 on teleconnections and Reviewer 1 on spatial trends).

The paragraph in Line 55 address explains better this rationale now.

'Increase in wave height (and wave energy) is primarily driven by an increase in surface wind energy (Wang et al. 2014). Wave heights have been increasing with upward trends in global sea-surface mean and extreme wind speeds across ocean basins from 1988 to 2011 (Young et al. 2011, 2012; Zheng et al. 2016). However, interannual climate patterns, like the Southern Annular Mode (SAM) or the Pacific Decadal Oscillation drive decadal basin-scale trends in wind stress throughout the global Ocean (Yang et al. 2007; Merrifield et al. 2012). Consequently, the global wave climate varies as a response to these atmospheric changes (Stopa & Cheung 2014), and changes in waves have been occurring associated with interannual climate variations and teleconnections in both hemispheres....'

L135: What are "known relationships" in this context?

Authors:

The text has been rewritten as:

Lines 221-223: *'...For this reason, we study spatial SST teleconnections with WP first by calculating the correlation of WP with two SST-based climate indices. Climate indices are ...'*

The comparison with known relationships with other wave parameters, e.g. (Hemer et al. 2010; Izaguirre et al. 2011; Stopa & Cheung 2014; Reguero et al. 2015), are included in the Discussion to compare with the patterns we find in this work.

L141: AMO is not defined.

Authors: Corrected.

L156-164: I don't know is this is a practice of Nature, but stating conclusions and interpretation on figures captions is not exactly very correct.

Authors:

All the Figure captions have been revised and modified. The discussion and conclusion statements have been removed from Figure captions.

L169: wave swell or just wave generation areas?

Authors:

Corrected to '*wave generation areas*'. The word '*global*' in the sentence has also been deleted.

L168: "Outside their generation zone, swells are not significantly affected by surface winds.". And? What do you exactly want to convey here?

Authors:

Because swell conditions are the most energetic sea states, this is just a remark to understand how critical are changes in these areas. We have clarified this in the text:

L250-252: 'More than 90% of the storm-wave maxima is generated in the generation areas in the extratropical sub-basins. Outside these generation areas, swells are not significantly affected by surface winds (Alves 2006). Therefore, the teleconnections of regional SSTs with the WP in these generation areas are particularly useful to explain how SST warming can increase WP globally because the local contribution of winds to WP can be considered to be negligible outside them.'

L182: TATL is defined on figure's 4 caption, and should be in the text body.

Authors: Corrected, also for the other regions.

L193: warming SST or increasing SST?

Authors: Corrected to *increasing SST*.

L197: larger or higher?

Authors: Corrected to *higher*.

L206: many is not exactly a very correct dimension... Several, maybe.

Authors: Amended.

L210: storm belts? What is this?

Authors:

Storm belt refers to an area of the ocean in which storms are frequent. This is the term that Hemer et al (2010) and other works use, also referred to 'wind belts' in (Young et al. 2011). We have opted to maintain the term.

L247: Why 5%? Where is this shown? Isn't this speculative?

Authors:

This is the value calculated from the regression line. This has been clarified in the new writing. It is also now represented in Figure 5-a in the main manuscript. Figure 5 (previously in the supplementary) has been added to the main body of the article as suggested by Reviewer 2.

The value has been adapted to provide the % increase taken the WP in 2008 as reference, and now is explained how it is calculated.

Line 199: 'GWP has been increasing linearly with SST at a rate of $1.8 \cdot 10^5$ kilowatts m^{-1} year $^{-1}$ over the global ocean, as a result of sea-surface warming (Figure 5-a): $GWP \cdot 10^{-5} = 1.804 \Delta SST + 2.606$ (normalized root mean square error = 12%).'

L253: 0.1°C increase leads to 6.4% increase in the global wave energy flux... Therefore 2.0° in the RCP8.5 will lead to an increase of...?

Authors:

This point is specifically now discussed in a new paragraph in Lines 296-306:

“ Ocean warming has been greatest near the surface (IPCC 2013). SST has warmed by 0.06 to 0.1°C per decade (Figure 2) but climate projections indicate that this trend will be stronger in tropical and northern-hemisphere subtropical regions in the future. Based on a linear regression between the global time series of GWP and SST (Figure 5-a), a 0.06-0.1°C increase per decade represents a GWP increase between 4.2 and 6.9% per decade (calculated with respect to the mean WP during 1948-2008). Projections of warming in the top 100 meters of the ocean range from approximately 0.6°C (RCP2.6) to 2.0°C (RCP8.5) by the end of the century, relative to 1986–2005 (IPCC 2013). For these projected increases in SST, the regression of the time series gives increases in GWP between 32 to 122% (relative to GWP during 1986–2005; an equivalent change in wave heights would be 16% to 48%, leaving other factors unchanged). These estimates can be compared with projected future changes in the wave climate. ‘

Although we compare these estimates with other works in subsequent paragraphs, we caution about the caveats of using a direct extrapolation based on empirical data. We refer the Reviewer to the extended reply to Reviewer 2 on this regard, in particular about (1) the comparison with the projected changes in Hs and wave energy flux in other works and (2) limitations of a direct extrapolation of the historical regression into the century.

L394: significant wave height.

Authors: Corrected

L398: Statement starting with “For swell...” is not correct. What are swell dominated sea states?

Authors: Corrected to ‘For swell sea states, ‘

L415: Why only until 2008?

Authors:

The Global Ocean Waves Reanalysis only covers to 2008, in its current version.

L416: Why didn't you use the GOW2 data set?

Authors:

We did not use GOW2 for several reasons:

First and foremost, GOW provides longer-time series that are more adequate to characterize long-term changes. We have calculated trends on the different eras of the reanalysis (new Figure 4 and supplementary material) and also by intervals of 20- and 10-years, and found significant differences in trend detection, mostly associated to interannual effects on the time series (see responses to Reviewer 1).

Second, we wanted to use an independent dataset. For this reason, we use the RaA13 hindcast data that uses a similar parameterization and wind forcing than GOW2.0 but is developed by an independent team of researchers.

Third, at the date of the time of this analysis, the data was not available yet and had not been published. Furthermore, the climatology of this dataset has not yet been validated, while the original GOW's WP has been validated with buoys specifically.

L428-429: Did you use Tm from altimetry measurements?

Authors:

We use the T_m from GOW, interpolated for the time steps and locations of the altimetry measurements. This has been clarified specifically in the methods because it was not properly acknowledged.

However, we also note that the WP depends weakly on T_m , as compared to the significant wave height ($\sim H_s^2$).

L438: "... different wind forcing data"... which are...?

Authors:

GOW uses NCEP-NCAR (Kalnay et al. 1996) while RaA13 uses the Climate Forecast System Reanalysis (Saha & Coauthors 2010). This clarification has been added to the Methods.

L444: ERA40 is not a reference to be stated. There are much more accurate products (starting with ERA-Interim).

Authors: Corrected.

REFERENCES

- Alves, J. (2006). Numerical modeling of ocean swell contributions to the global wind-wave climate. *Ocean Model.*, 11, 98–122.
- Burkey, J. (2006). A non-parametric monotonic trend test computing Mann-Kendall Tau, Tau-b, and Sen's Slope written in Mathworks-MATLAB implemented using matrix rotations.
- Cai, W., Borlace, S., Lengaigne, M., van Rensch, P., Collins, M., Vecchi, G., Timmermann, A., Santoso, A., McPhaden, M.J., Wu, L., England, M.H., Wang, G., Guilyardi, E. & Jin, F.-F. (2014). Increasing frequency of extreme El Niño events due to greenhouse warming. *Nat. Clim. Chang.*, 4, 111.
- Dodet, G., Bertin, X. & Taborda, R. (2010). Wave climate variability in the North-East Atlantic Ocean over the last six decades. *Ocean Model.*, 31.
- Fan, Y., Held, I.M., Lin, S.-J. & Wang, X.L. (2013). Ocean Warming Effect on Surface Gravity Wave Climate Change for the End of the Twenty-First Century. *J. Clim.*, 26, 6046–6066.
- Fan, Y., Lin, S.J., Griffies, S.M. & Hemer, M. a. (2014). Simulated global swell and wind-sea climate and their responses to anthropogenic climate change at the end of the twenty-first century. *J. Clim.*, 27, 3516–3536.
- Hemer, M.A., Church, J.A. & Hunter, J.R. (2010). Variability and trends in the directional wave climate of the Southern Hemisphere wave climate of the Southern Hemisphere. *Int. J. Climatol.*, 30, 475–491.
- Hemer, M. a. (2010). Historical trends in Southern Ocean storminess: Long-term variability of extreme wave heights at Cape Sorell, Tasmania. *Geophys. Res. Lett.*, 37, 1–5.
- Hemer, M. a., Fan, Y., Mori, N., Semedo, A. & Wang, X.L. (2013). Projected changes in wave climate from a multi-model ensemble. *Nat. Clim. Chang.*, 3, 471–476.
- IPCC. (2013). *Climate Change 2013: The Physical Science Basis. Contribution of Working Group I to the Fifth Assessment Report of the Intergovernmental Panel on Climate Change*. Cambridge University Press, Cambridge, United Kingdom and New York, NY, USA, 1535 pp.
- Izaguirre, C., Mendez, F.J., Menendez, M. & Losada, I.J. (2011). Global extreme wave height variability based on satellite data. *Geophys. Res. Lett.*, 38.
- Kalnay, E., Kanamitsu, M., Cistler, R., Collins, W., Deaven, D., Gandin, L., Iredell, M., Saha, S., White, G., Woolen, J., Zhu, Y., Chelliah, M., Ebisuzaki, W., Higgins, W., Janowiak, J., Mo, K.C., Ropelewski, C., Wang, J., Leetma, A., Reynolds, R., Jenne, R. & Joseph, D. (1996). The NCEP/NCAR Reanalysis Project. *Bull. Am. Meteorol. Soc.*, 77, 437–471.
- Mentaschi, L., Vousdoukas, M.I., Voukouvalas, E., Dosio, A. & Feyen, L. (2017). Global changes of extreme coastal wave energy fluxes triggered by intensified teleconnection patterns. *Geophys. Res. Lett.*, 44, 2416–2426.
- Merrifield, M.A., Thompson, P.R. & Lander, M. (2012). Multidecadal sea level anomalies and trends in the western tropical Pacific. *Geophys. Res. Lett.*, 39, n/a-n/a.
- Mínguez, R., Reguero, B.-. G., Luceño, a. & Méndez, F. ~J. J. (2011). Regression Models for Outlier Identification (Hurricanes and Typhoons) in Wave Hindcast Databases. *J. Atmos. Ocean. Technol.*, 29, 267–285.
- Mori, N., Yasuda, T., Mase, H., Tom, T. & Oku, Y. (2010). Projection of extreme wave climate under global warming. *Hydro. Res. Lett.*, 4, 14–19.
- Rasclé, N. & Ardhuin, F. (2013). A global wave parameter database for geophysical applications. Part 2: Model validation with improved source term parameterization. *Ocean Model.*, 70, 174–188.
- Reguero, B.G., Losada, I.J. & Méndez, F.J. (2015). A global wave power resource and its seasonal, interannual and long-term variability. *Appl. Energy*, 148, 366–380.
- Reguero, B.G., Menéndez, M., Méndez, F.J., Mínguez, R. & Losada, I.J. (2012). A Global Ocean Wave (GOW) calibrated reanalysis from 1948 onwards. *Coast. Eng.*, 65, 38–55.
- Saha, S. & Coauthors. (2010). The NCEP climate forecast system reanalysis. *Am. Meteorol. Soc. Bull.*, 91, 1015–1057.
- Shimura, T., Mori, N. & Mase, H. (2015). Future Projection of Ocean Wave Climate: Analysis of SST Impacts on Wave Climate Changes in the Western North Pacific. *J. Clim.*, 28, 3171–3190.
- Sterl, A. & Caires, S. (2005). Climatology, variability and extrema of ocean waves: The web-based KNMI/ERA-40 wave atlas. *Int. J. Climatol.*, 25, 963–977.

- Stopa, J.E. & Cheung, K.F. (2014). Periodicity and patterns of ocean wind and wave climate. *J. Geophys. Res. Ocean.*, 119, 5563–5584.
- Vermeer, M. & Rahmstorf, S. (2009). Global sea level linked to global temperature. *Pnas*, 2009, 1–6.
- Vinoth, J. & Young, I.R. (2010). Global Estimates of Extreme Wind Speed and Wave Height. *J. Clim.*, 24, 1647–1665.
- Wang, X.L., Feng, Y. & Swail, V.R. (2014). Changes in global ocean wave heights as projected using multimodel CMIP5 simulations. *Geophys. Res. Lett.*, 41, 1026–1034.
- Wang, X.L., Feng, Y. & Swail, V.R. (2015). Climate change signal and uncertainty in CMIP5-based projections of global ocean surface wave heights. *J. Geophys. Res. Ocean.*, 120, 3859–3871.
- Wang, X.L. & Swail, V.R. (2001). Changes of extreme Wave Heights in northern Hemisphere Oceans and related atmospheric circulation regimes. *J. Clim.*, 14, 2204–2221.
- Wang, X.L. & Swail, V.R. (2002). Trends of Atlantic wave extremes as simulated in a 40-yr wave hindcast using kinematically reanalyzed wind fields. *J. Clim.*, 15, 1020–1035.
- Wang, X.L. & Swail, V.R. (2006). Climate change signal and uncertainty in projections of ocean wave heights. *Clim. Dyn.*, 26, 109–126.
- Wang, X.L., Swail, V.R., Zwiers, F.W., Zhang, X. & Feng, Y. (2009). Detection of external influence on trends of atmospheric storminess and northern oceans wave heights. *Clim. Dyn.*, 32, 189–203.
- Yang, X.-Y., Huang, R.X. & Wang, D.X. (2007). Decadal Changes of Wind Stress over the Southern Ocean Associated with Antarctic Ozone Depletion. *J. Clim.*, 20, 3395–3410.
- Young, I.R., Vinoth, J., Zieger, S. & Babanin, a. V. (2012). Investigation of trends in extreme value wave height and wind speed. *J. Geophys. Res. Ocean.*, 117, 1–13.
- Young, I.R., Zieger, S. & Babanin, a V. (2011). Global trends in wind speed and wave height. *Science (80-)*, 332, 451–455.
- Zheng, C.W., Pan, J. & Li, C.Y. (2016). Global oceanic wind speed trends. *Ocean Coast. Manag.*, 129, 15–24.

Reviewer #1 (Remarks to the Author):

I think this manuscript is now acceptable for publication. I only have the following minor comments:

Line 132: change "(0.66) time series" to "(0.66) GWP time series"

L147: change "; has" to ", has"

Line 201: delete the first "the" in "the not only the")

Reviewer #3 (Remarks to the Author):

Second review iteration of the NCOMMS manuscript "Uncovering a new global signal of climate change: evidences of a stronger global wave climate associated to ocean warming" by Reguero et al.

I am afraid I still have concerns about this manuscript. Fair to say that the manuscript has been substantially improved. Nevertheless I am not fully convinced that the merits of the study are worthy of publication in such a high profile journal. Usually I do not accept reviewing manuscripts that I have previously rejected. Nevertheless, attending to the journal, and to the fact that the editors let the manuscript move through, with two major revisions and a reject, I have made an exception. For that matter I have revised the manuscript once more.

Despite the fact that I am not fully convinced, leave the final decision to the editors, taking into account the author's work and the two other reviewers opinion. I am adding comments to the rebuttal and to the revised manuscript below.

Comments to the rebuttal:

I am not satisfied with your answer in the rebuttal to my query "L416: Why didn't you use the GOW2 data set?"

It seems that you are using a substandard data set, after your group already produced, evaluated, and published the results of a considerably better product (GOW2). Let's face it, The GOW wave data set is presently only comparable to ERA40, and who would use ERA40 for such a detailed study?

GOW2 has been evaluated and the results published.

GOW2 has in fact a shorter time series, but how important is that? GOW2 is close to 40 years long.

The need for an independent data set for comparison is also not sound, since there are several other products that can be used for that.

In my opinion GOW is an outdated dataset, with substandard quality compared to the wave reanalysis and hindcasts, of higher resolution, available, including GOW2.

I leave this point to the editors consideration, but I consider this point, in view of the high profile of Nature Climate Change, a rather questionable point.

Comments to the revised manuscript:

The manuscript is now considerably longer. Does it still comply with the journal's requirements?

L19: "Ocean warming has been changing waves globally". Said like this it seems that the only cause of changes in the global wave climate is the recent warming of the ocean. Actually the manuscript is full of this type of grandiloquent conclusions, which are at least, for lack of a better word, exaggerated or lack support either from the results or from previous studies.

L30: Should there be a section here? Main body? Introduction?

L31/L35: replace in with at.

L34: Replace is with are.

L37: Isn't ocean warming part of climate change?

L43-44: Sentence starting with "Wave heights..." is confusing. Consider re-writing.

L55: Replace height with heights.

L56: Increasing upward trends is redundant.

L69: Replace one with a.

L71: Add reference after records.

L91: Again, the use of could shows little confidence in what you are stating. Replace with can. Why potentially? And why (L92) can WP better characterize the long term behaviour of the global wave conditions. How do you back this statement, including with references?

L108: I am afraid I still don't understand this statement that "WP represents the temporal variations of energy transferred from the atmosphere to the ocean surface over cumulative periods of time. Why is this? Provide a couple of sentences and an educated reference explaining this. Why is it a better indicator (L109) than other parameters?

L112: Why does it aggregate information from mean and extreme conditions? Of course it does. Do you mean it is amplified in extreme wave height conditions to the second order?

L114-L116: Again, how does WP represent the accumulated wave energy over a certain period of time?

L118? You calculate the time series? Or do you analyse them?

L119: Add previously after been.

L122: GWP already defined previously.

L125: You cannot validate model output. At the most you evaluate it. And certainly not by comparing it to another hindcast.

L137: Temperature warming? Replace increases with has increased.

L142: You have a problem with the Southern Ocean. Basically you consider everything as Southern Ocean south of mid-latitudes. According to the International Hydrographic Organisation (Limits of Oceans and Seas (2002)), the Southern Ocean starts at 60°S. This is of course a political delimitation, but you seem to mix the southern parts of the Pacific, Atlantic and Indian Oceans all in the Southern Ocean. Some clarification is needed. Maybe stating that you do that for convenience.

L152: It doesn't have "some of the largest extreme wave heights"; it has the most extreme wave heights, or the highest waves in the global ocean.

L154: Don't understand the sentence starting with "This suggests..." Suggests? And how can you relate the extreme wave heights and WP. They are different parameters (although the latter is derived from the former to the second order). What is the relation here? What exactly do you want to convey here?

L158: Could? (Entire paragraph from L157 to L163 is a bit confusing.)

L165: What is ERSSTv3b?

L166: What is OISTT?

L183: Add reference after theory.

L190: Put Figure 4-a,b inside curly brackets.

L192: Replace colon by semi-colon before see.

L200: sentence in this line is a bit confusing.

L207: warming of what?

L213: consider revising sentence starting with "An analysis..."

L223-224: Replace to allow with allowing.

L228: Why are they "the most"? (See comment on L19 above.)

L231-232: Isn't dynamic atmospheric factors, in this context, what we call atmospheric bridging?

L234: Suggest revising sentence starting with "For WP..."

L245-246: Alves (2006) does not identify wave climate regions; just regions. That is was not his purpose nor are those his conclusions.

L246: Don't understand the 30S-30N limits in Alves (2006) context.

L247: Who says that "more than 90% of the storm wave maxima are generated in the extratropical sub-basins"?

L259: Where is the tropical Pacific? Intertropical? Equatorial? Why/how equivalent?

L265: Consider revising sentence starting with "This pattern...".

L281: Replace north Pacific with North Pacific; do the same for the remainder of the manuscript, also for other oceans.

L283: Are you sure they don't receive swells for other basins?

L286: erase keep spatial; replace agreement with agree.

L288: What are milder climate conditions? Aren't you referring to weather?

L291: See comment on L142 regarding the Southern Ocean.

L299: Add reference after future.

L305: can or are?

L308: Consider revising sentence starting with "The average...".

L316: Which estimates? Hemer et al. 2013 presents an opportunity ensemble, based on several dynamical and statistical projections, with mixed AR4 scenarios. Careful should be taken when comparing their results, and that should be mentioned in the text, maybe referencing the partial studies that gave rise to this ensemble.

L395: Add an before increase.

L398: Why is it a new indicator? Wave energy flux is a sea state parameter. Any wave or sea state parameter is an indicator of climate change. Wave periods vary with climate change, mean wave direction varies with climate change, wave steepness varies with climate change, so on and so forth. To an extent that all wave parameters (since waves are part of the climate system) can be seen as "new" indicators of climate change. What you can say, in my opinion, is that the impact of climate warming on the wave climate can also be seen or is reflected in the energy transported by the waves.

L432: Only for swell?

L476: Replace is with are.

L460: Essentially unchanged? How is a reanalysis "essentially unchanged"?

Point by point responses to referees

Reviewers' comments:

Reviewer #1 (Remarks to the Author):

I think this manuscript is now acceptable for publication. I only have the following minor comments:

Line 132: change "(0.66) time series" to "(0.66) GWP time series"

L147: change "; has" to ", has"

Line 201: delete the first "the" in "the not only the")

Authors:

The three points have been attended and marked in blue in the revised version of the manuscript.

Reviewer #3 (Remarks to the Author):

Second review iteration of the NCOMMS manuscript "Uncovering a new global signal of climate change: evidences of a stronger global wave climate associated to ocean warming" by Reguero et al.

I am afraid I still have concerns about this manuscript. Fair to say that the manuscript has been substantially improved. Nevertheless, I am not fully convinced that the merits of the study are worthy of publication in such a high profile journal. Usually I do not accept reviewing manuscripts that I have previously rejected. Nevertheless, attending to the journal, and to the fact that the editors let the manuscript move through, with two major revisions and a reject, I have made an exception. For that matter I have revised the manuscript once more.

Despite the fact that I am not fully convinced, leave the final decision to the editors, taking into account the author's work and the two other reviewers' opinion. I am adding comments to the rebuttal and to the revised manuscript below.

Authors:

We appreciate the reviewer's input and time, particularly for reconsidering the manuscript and providing another revision.

We note that we have attended the points raised and added the required new dataset. The analyses have been updated accordingly.

Comments to the rebuttal:

I am not satisfied with your answer in the rebuttal to my query "L416: Why didn't you use the GOW2 data set?"

It seems that you are using a substandard data set, after your group already produced, evaluated, and published the results of a considerably better product (GOW2). Let's face it, The GOW wave data set is presently only comparable to ERA40, and who would use ERA40 for such a detailed study? GOW2 has been evaluated and the results published. GOW2 has in fact a shorter time series, but how important is that? GOW2 is close to 40 years long. The need for an independent data set for comparison is also not sound, since there are several other products that can be used for that.

In my opinion GOW is an outdated dataset, with substandard quality compared to the wave reanalysis and hindcasts, of higher resolution, available, including GOW2. I leave this point to the editors consideration, but I consider this point, in view of the high profile of Nature Climate Change, a rather questionable point.

Authors:

The analysis has been updated with the latest version of the global wave dataset GOW2 (hereafter referred to as GOW-CFSR).

We note that this hindcast is not an update of the original GOW but an independent dataset, published by (Perez et al., 2017), and presents different wind forcing, resolution and model set up. The GOW-CFSR provides global wave data at 0.5 degrees globally, uses the improved parameterization developed by (Ardhuin et al., 2010) and implemented and validated in the second global wave hindcast considered originally in this manuscript, i.e. RaA13, which covers the period 1994-2013 (Rascle and Ardhuin, 2013). While the original GOW was calibrated with satellite-derived wave heights through a directional correction, the GOW-CFSR does not include any correction to the numerical outputs.

GOW-CFSR uses the same wind forcing that RaA13, i.e. the NCEP Climate Forecast System Reanalysis. However, it provides the main advantage of offering wave data up to the year 2017 (RaA13 ends in 2013). The GOW-CFSR constitutes the third numerical global wave reanalysis used in this study to estimate the Global Wave Power (GWP). The analysis has been repeated with GOW-CFSR and the results are consistent with the other wave data sources.

Each of the three wave datasets present different characteristics: GOW provides the longest and a calibrated climatology; RaA13 shows the best fit with satellite-altimetry data; and the GOW-CFSR the most updated coverage adding data for the period from 2013 to 2017. We have included a comparison of these three numerically derived datasets with the global time series of Global Wave Power (GWP) calculated from the satellite altimetry. This analysis has been included in the Supplementary Information (see Supplementary Fig.1). Both CFSR datasets provide a better agreement with the satellite-derived GWP than the original GOW-NCEP.

We have updated Figures 2 and 3 with the new GOW-CFSR time series. It is important to note that the addition of the GOW-CFSR now includes two severe ENSO events in Figure 3, which further confirms the behavior found for the original time series.

Figure 2. Historical variability in oceanographic forcing. (a) GWP time series from GOW (black: GOW-NCEP, presented in Figure 1; and grey: GOW-CFSR), RaA13 (blue points) and satellite altimetry (red

points); (b) global SST time series from ERSSTv3b (blue) and OISST (red); and (c) non-autocorrelated residuals of WP from GOW (black and grey) and SST from ERSSTv3b (blue) and OISST (red). The vertical dashed lines indicate the beginning of the era in which wave height has been measured with satellites.

Figure 3. Global seasonal variability in the seasonal GWP (blue: GOW-NCEP; black: GOW-CFSR) and SST anomalies (red). Years corresponding to strong El Niño events, that is, ones with an Oceanic Niño Index value exceeding 1.5, are annotated in gray and overlaid on the graph.

The correspondence between the RaA13 and GOW-CFSR with the GWP derived from satellite altimetry shows a good agreement (see Figure below). The CFSR wave data is only used after the year 1994 because previous studies have found large errors and biases in wind speeds and wave climatology in high latitudes of the Southern Hemisphere, both in mean and extreme values, due to an incorrect wind data assimilation in the CFSR until the SSM-I data started to be included in 1994 (see (Arduin et al., 2011; Rasclé and Arduin, 2013; Saha and Coauthors, 2010)). It is possible that a correction of the CFSR wind speed histogram may be enough to correct the biases on wave parameters, but both the RaA13 and GOW-CFSR are uncorrected before the year 1994, as noted and studied in (Rasclé and Arduin, 2013). The climatology GOW-CFSR was validated in (Perez et al., 2017) for the period 1992-2015 (see Fig 3 in (Perez et al., 2017)). However, such validation shows up to 0.5m in bias of the mean significant wave heights in high latitudes of the Southern Hemisphere (the most energetic region in terms of WP).

Supplementary Figure 1. Comparison of the GWP calculated from satellite altimetry (x-axis) and the numerically derived GWP (y-axis) for the wave datasets used in the analysis.

The diagnostic statistics for comparing hindcast performance (y) with respect to the satellite altimetry GWP (x) are calculated as follows: $Bias = \bar{x} - \bar{y}$; Residual Scatter Index: $SI = \frac{RMSE}{\bar{x}}$, where $RMSE = \sqrt{\frac{1}{n_d} \sum_{i=1}^{n_d} (x_i - y_i)^2}$; and R-square as the Coefficient of Determination of the linear regression between x and y.

Comments to the revised manuscript:

The manuscript is now considerably longer. Does it still comply with the journal's requirements?

Authors:

The manuscript has been extended in the Methods section, following indications by the Editor and the reviewers. The rest of the manuscript complies with the extension limitation indicated by the journal. However, we note we are over the limit number of references (70 is given as a reference), but we consider they are necessary to include.

L19: "Ocean warming has been changing waves globally". Said like this it seems that the only cause of changes in the global wave climate is the recent warming of the ocean. Actually the manuscript is full of

this type of grandiloquent conclusions, which are at least, for lack of a better word, exaggerated or lack support either from the results or from previous studies.

Authors:

This sentence did not aim to convey that global warming is the only factor affecting variability in waves, but, rather, a contributing factor (as pointed out by other reviewers in the previous revision, and now better explained in the Introduction). In case the lack of context in the abstract is misleading, we have rewritten it to avoid further misinterpretation. We now indicate that is one of the factors affecting waves globally.

The paragraph in Line 43 introduces the identified long-term changes in wave parameters from different sources. The transition to the significant wave changes that occur associated with interannual variability is made subsequently (e.g. Lines 57-69). These two paragraphs take the reader to the focus of the paper, the long-term changes in GWP and its interannual variations in relation with the SST warming and its spatio-temporal variability.

We have revised the manuscript to avoid other misleading contexts and misinterpretations and rewritten this sentence in the abstract. Also, as pointed and requested by one of the other reviewers, we explicitly expressed the influence of climate change on other wave parameters, for example in Line 38-39: *Waves result from the interaction between the atmosphere and the ocean and therefore are affected by climate change and ocean warming (Fan et al., 2013; Wang et al., 2009; Wang and Swail, 2006, 2001; Young et al., 2011).*

L30: Should there be a section here? Main body? Introduction?

Authors:

The journal guidelines specifically indicate that there should not be an Introduction headline. '*... Section headings should be used and subheadings may appear in 'Results'. Avoid 'Introduction' as a heading.*'

L31/L35: replace in with at.

Authors: Corrected (at the coast).

L34: Replace is with are.

Authors: Corrected.

L37: Isn't ocean warming part of climate change?

Authors: Corrected. We have left only the term 'climate change'.

L43-44: Sentence starting with "Wave heights..." is confusing. Consider re-writing.

Authors: The sentence has been rewritten.

L55: Replace height with heights.

Authors: Corrected

L56: Increasing upward trends is redundant.

Authors:

The sentence has been rewritten as:

L59: *Wave heights have been increasing associated to upward trends in global sea-surface mean and extreme wind speeds across ocean basins from 1988 to 2011*

L69: Replace one with a.

Authors: Corrected

L71: Add reference after records.

Authors:

We have added the reference to the IPCC chapter that describes the different data sources and changes in SST and other temperature variables. We have also included the reference in the Methods section, when describing the SST: (Hartmann et al., 2013), Chapter 2, page 190 onwards.

L91: Again, the use of could shows little confidence in what you are stating. Replace with can. Why potentially? And why (L92) can WP better characterize the long term behavior of the global wave conditions. How do you back this statement, including with references?

Authors:

The original use of 'could' instead of 'can' did not aim to be an affirmative statement but, instead, to present our hypothesis and introduce the focus of the paper that is developed in the subsequent sections. However, attending to the journal guidelines that specify: *'the last paragraph contains a brief summary of both the results and the conclusions (written in present tense)'*, we changed this tense as recommended.

We have opted to follow the journal's guidelines and leave this summary paragraph without references. The references are incorporated in the corresponding argumentation and sections (Results and Discussion sections).

L108: I am afraid I still don't understand this statement that "WP represents the temporal variations of energy transferred from the atmosphere to the ocean surface over cumulative periods of time. Why is this? Provide a couple of sentences and an educated reference explaining this. Why is it a better indicator (L109) than other parameters?

Authors:

The paragraph has been rewritten to clarify this point.

WP is a useful variable to study the long-term behavior of the global wave climate because it aggregates information on wave heights and periods and is cumulative over time. For example, the mean annual significant wave height does not include information of the history of the wave conditions over a year: two places may present a similar mean H_s but with different storm record, for ex one location with generally small conditions, e.g. 1m, but with large events, e.g. 10m, as compared with other location with less variability but larger average conditions throughout, e.g. 2-3m. While the H_s may be similar in both, the WP will be drastically different.

The new paragraph has been rewritten as follows (L109):

WP is the transport of energy by waves and represents the temporal variations of energy transferred from the atmosphere to the ocean surface motion over cumulative periods of time. WP can be a better indicator of long-term behavior of the global wave conditions than other parameters, such as the wave heights, because, first, it includes information on significant wave heights (the mean value of the largest third of the wave heights during typically 1 hour, H_s) but also periods (T) (see Methods). Second, being proportional to $H_s^2 T$, WP aggregates information of both mean and extreme conditions but increases proportionally more with the latter because it is amplified in extreme wave height conditions to the second order (i.e., high percentiles of wave heights and periods, which have shown stronger trends than mean

conditions). Third, WP can represent the accumulated wave energy over periods of time, for example months, seasons and years, unlike other wave climate parameters, such as wave heights, that must be averaged over the same periods of time. WP can therefore represent variations in the wave climate not captured by other parameters (e.g. an annual mean wave height). For example, two locations can have similar average values of wave heights but drastically different energy over a year depending on the storm activity.

L112: Why does it aggregate information from mean and extreme conditions? Of course it does. Do you mean it is amplified in extreme wave height conditions to the second order?

Authors:

The writing suggestion has been incorporated in the revised paragraph. See previous response.

L114-L116: Again, how does WP represent the accumulated wave energy over a certain period of time?

Authors: See previous response. This has been clarified in the revised text.

L118? You calculate the time series? Or do you analyse them?

Authors:

The writing has been revised for precision. We calculate the global time series and then analyze them.

L119: Add previously after been.

Authors: Added

L122: GWP already defined previously.

Authors: Corrected

L125: You cannot validate model output. At the most you evaluate it. And certainly not by comparing it to another hindcast.

Authors: We have used 'support' instead. The text in the Methods section has also been revised.

L137: Temperature warming? Replace increases with has increased.

Authors: Corrected.

L142: You have a problem with the Southern Ocean. Basically you consider everything as Southern Ocean south of mid-latitudes. According to the International Hydrographic Organization (Limits of Oceans and Seas (2002)), the Southern Ocean starts at 60°S. This is of course a political delimitation, but you seem to mix the southern parts of the Pacific, Atlantic and Indian Oceans all in the Southern Ocean. Some clarification is needed. Maybe stating that you do that for convenience.

Authors:

This is an important point. Although it was defined in the Figure caption, we have opted to add a more detailed description of the spatial delimitation of the Southern Ocean in the Methods Section.

We refer to the 'Southern Ocean' as the region south of the 40 degrees latitudinal limit. This delimitation is based on the wave power climatology and the storm activity in the southern Hemisphere that shows a continuous area of high energy beyond that latitude (see Figure below). Additionally, this delimitation is

also more consistent with the southern regions studied by Alves (2006) to understand the swell generation zones, which is subsequently used to explore the spatial teleconnections.

This delimitation has been explained in the text in the Methods section when explaining how the regional time series are calculated, and in the Figure where we refer to the Southern Ocean.

L152: It doesn't have "some of the largest extreme wave heights"; it has the most extreme wave heights, or the highest waves in the global ocean.

Authors:

The text has been corrected. However, there are comparable regions in the Pacific and the Atlantic for the 100-year significant wave height (see Supplementary Fig 3). We prefer to maintain the expression *'has some of the most extreme wave heights on the planet'*.

L154: Don't understand the sentence starting with "This suggests..." Suggests? And how can you relate the extreme wave heights and WP. They are different parameters (although the latter is derived from the former to the second order). What is the relation here? What exactly do you want to convey here?

Authors:

The sentence has been rewritten. The point addressed in this paragraph indicates that the WP represents characteristics of the wave climate not captured in other parameters, like the extreme significant wave heights.

L158: Could? (Entire paragraph from L157 to L163 is a bit confusing.)

Authors:

We have opted to maintain 'could' because we are describing the hypothesis that we will be studying in the subsequent analysis. However, the text in the paragraph has been modified.

L165: What is ERSSTv3b?

Authors:

ERSSTv3b is the extended reconstructed SST dataset. The term is introduced in the preceding paragraph and further described in the Methods.

L166: What is OISTT?

Authors:

OISTT is the NOAA's optimum 162 interpolation SST, introduced in the foregoing paragraph.

L183: Add reference after theory.

Authors: Added.

L190: Put Figure 4-a,b inside curly brackets.

Authors: Corrected

L192: Replace colon by semi-colon before see.

Authors: Corrected

L200: sentence in this line is a bit confusing.

Authors: It has been simplified to only introduce the regression.

L207: warming of what?

Authors: Corrected

L213: consider revising sentence starting with "An analysis..."

Authors: Rewritten

L223-224: Replace to allow with allowing.

Authors: Corrected

L228: Why are they "the most"? (See comment on L19 above.)

Authors:

These are the two SST-based indices with strongest correlations with WP, based on (Reguero et al., 2015). We have added the reference in the text.

L231-232: Isn't dynamic atmospheric factors, in this context, what we call atmospheric bridging?

Authors:

Yes, it has been corrected.

This reference refers to the delayed ocean transport mechanism by which ENSO provides an additional heat supply favorable for the formation of strong hurricanes over the eastern North Pacific.

We have added an additional reference, Alexander et al 2002, which explains the atmospheric bridge and the response of the global ocean conditions to sea surface temperature anomalies in the equatorial Pacific.

L234: Suggest revising sentence starting with "For WP..."

Authors:

The sentence has been rewritten.

L245-246: Alves (2006) does not identify wave climate regions; just regions. That is was not his purpose nor are those his conclusions.

Authors:

This is correct, the text has been rewritten. The text in the Methods has also been revised for precision.

We use the same oceanic areas that were used for studying the contribution of ocean swell to the global wind-wave climate in (Alves, 2006) because (i) outside these generation areas, swells are not significantly affected by surface winds (Alves, 2006). and (ii) they also represent different wave climate types (e.g. Rueda et al., 2017).

L246: Don't understand the 30S-30N limits in Alves (2006) context.

Authors:

Alves defines these regions by the 30S-30N limits. The sentence has been rewritten for further clarification.

Fig. 2. Selected swell generation areas: tropical north Indian (TNIO), tropical western North Pacific (TWNP), tropical eastern North Pacific (TENP), tropical North Atlantic (TNAO), tropical south Indian (TSIO), tropical western south Pacific (TWSP), tropical eastern south Pacific (TESP), tropical south Atlantic (TSAO), extratropical North Pacific (ETNP), extratropical North Atlantic (ETNA), extratropical south Indian (ETSI), extratropical south Pacific (ETSP) and extratropical south Atlantic (ETSA).

L247: Who says that “more than 90% of the storm wave maxima are generated in the extratropical sub-basins”?

Authors:

Alves (2006), in page 104, indicates that:

‘Data in Table 1 indicate that more than 90% of all extratropical storm maxima NSMAX around the globe are contained within selected discrete extratropical areas. Similarly, the tropical storm areas contain more than 95% of tropical storm maxima in the chosen storm database. Table 1 further reveals that two selected areas, TSAO and TESP, are nearly free of storms of any kind’

L259: Where is the tropical Pacific? Intertropical? Equatorial? Why/how equivalent?

Authors:

The tropical Pacific in this study is defined by the Pacific basin within the 30N and 30S. However, a warming in this region is dominated by ENSO events, which is why we find similar patterns than for the Niño index. The sentence has been rewritten.

L265: Consider revising sentence starting with “This pattern...”.

Authors:

The sentence has been rewritten.

L281: Replace north Pacific with North Pacific; do the same for the remainder of the manuscript, also for other oceans.

Authors:

Corrected, for both ‘North’ and ‘South’.

L283: Are you sure they don’t receive swells for other basins?

Authors:

The expression was not correct, it has been rewritten to be more precise and consistent with Alves (2006)

L286: erase keep spatial; replace agreement with agree.

Authors: Corrected.

L288: What are milder climate conditions? Aren’t you referring to weather?

Authors: Corrected. We have used ‘calm conditions’

L291: See comment on L142 regarding the Southern Ocean.

Authors:

The spatial definition has been included in the Methods and when first referring to the Southern Ocean.

L299: Add reference after future.

Authors: IPCC (2013) added.

L305: can or are?

Authors: ‘are comparable to’, the sentence has also been added to the next paragraph.

L308: Consider revising sentence starting with “The average...”.

Authors: The sentence has been revised and rewritten for clarity.

L316: Which estimates? Hemer et al. 2013 presents an opportunity ensemble, based on several dynamical and statistical projections, with mixed AR4 scenarios. Careful should be taken when comparing their results, and that should be mentioned in the text, maybe referencing the partial studies that gave rise to this ensemble.

Authors: This is an important remark. It has been indicated in the text.

L395: Add an before increase.

Authors: Corrected.

L398: Why is it a new indicator? Wave energy flux is a sea state parameter. Any wave or sea state parameter is an indicator of climate change. Wave periods vary with climate change, mean wave direction varies with climate change, wave steepness varies with climate change, so on and so forth. To an extent that all wave parameters (since waves are part of the climate system) can be seen as “new” indicators of climate change. What you can say, in my opinion, is that the impact of climate warming on the wave climate can also be seen or is reflected in the energy transported by the waves.

Authors: The expression has been changed to the suggested sentence.

L432: Only for swell?

Authors:

Classically, the significant wave height H_s can be estimated from a wave-by-wave analysis and calculated as the mean of the top 1/3 waves in a given record ($H_{1/3}$). However, it is more often estimated from the variance of the record or the integral of the variance in the spectrum, denoted as H_{m0} . The agreement between significant wave height estimates based on spectral moments (H_{m0}) versus zero-crossing analysis ($H_{1/3}$ or H_s) is linked to the underlying narrow band assumption. A divergence from this theory occurs as spectral width increases with changes in the wave field. Therefore, H_s can be approximated to H_{m0} for swell-type sea states and are more adequate for deep water conditions, while this assumption will be inaccurate for shallow waters where the spectral shape is substantially changed, e.g. (Vanderer et al., 2008).

L476: Replace is with are.

Authors: Corrected

L460: Essentially unchanged? How is a reanalysis “essentially unchanged”?

Authors:

The expression has been corrected.

We were using the same expression than the authors in Kistler et al (2001). The NCEP–NCAR 50-Year Reanalysis: Monthly Means CD-ROM and Documentation:

*‘We have emphasized that although the NCEP– NCAR reanalysis system **was essentially unchanged** during the more than 50 years processed, there were two major changes in the observing system. The first took place during 1948–57, when the upper-air network was being established, and the second in 1979 when the global operational use of satellite soundings was introduced. The introduction of satellite data in 1979 resulted in a significant change in the climatology, especially above 200 hPa and south of 50°S, suggesting that the climatology based on the years 1979–present day is most reliable.’*

We have clarified these changes in the revised text and added more precise information. The paragraph (L530) has been rewritten as:

‘ The NCEP/NCAR reanalysis assimilation system ⁶¹ was unchanged during the reanalysis period to eliminate climate jumps, although the reanalysis is still affected by changes in the observing data ^{70,75}. There were two changes in the observing data: before and after 1957 (eras I and II) when the upper-air network was being established and that mainly affected the southern hemisphere ⁶²; and the introduction of the global operational use of satellite soundings in 1979. It has been therefore recommended that the periods before and after 1979 are studied

independently because the climatology based on the years 1979–present day is most reliable⁷⁵. Following this recommendation, our analysis separates the full historical period (1948 onwards) and the satellite era (1979 onwards). The results for the eras not included in the main manuscript can be found in the Supplementary Information. ‘

References in this document

- Alves, J., 2006. Numerical modeling of ocean swell contributions to the global wind-wave climate. *Ocean Model.* 11, 98–122.
- Ardhuin, F., Hanafin, J., Quilfen, Y., Chapron, B., Queffelec, P., Obrebski, M., Sienkiewicz, J., Vandemark, D., 2011. Calibration of the IOWAGA global wave hindcast (1991–2011) using ECMWF and CFSR winds, in: *Proceedings, 12th International Workshop of Wave Hindcasting and Forecasting, Hawaii.*
- Ardhuin, F., Rogers, E., Babanin, A. V., Filipot, J.-F., Magne, R., Roland, A., van der Westhuysen, A., Queffelec, P., Lefevre, J.-M., Aouf, L., Collard, F., 2010. Semiempirical Dissipation Source Functions for Ocean Waves. Part I: Definition, Calibration, and Validation. *J. Phys. Oceanogr.* 40, 1917–1941. <https://doi.org/10.1175/2010JPO4324.1>
- Hartmann, D.L., Klein Tank, A.M.G., Rusticucci, M., Alexander, L. V., Brönnimann, S., Charabi, Y.A.R., Dentener, F.J., Dlugokencky, E.J., Easterling, D.R., Kaplan, A., Soden, B.J., Thorne, P.W., Wild, M., Zhai, P., 2013. Observations: Atmosphere and surface. *Clim. Chang. 2013 Phys. Sci. Basis Work. Gr. I Contrib. to Fifth Assess. Rep. Intergov. Panel Clim. Chang.* 9781107057, 159–254. <https://doi.org/10.1017/CBO9781107415324.008>
- Perez, J., Menendez, M., Losada, I.J., 2017. GOW2: A global wave hindcast for coastal applications. *Coast. Eng.* 124, 1–11. <https://doi.org/http://doi.org/10.1016/j.coastaleng.2017.03.005>
- Raschle, N., Ardhuin, F., 2013. A global wave parameter database for geophysical applications. Part 2: Model validation with improved source term parameterization. *Ocean Model.* 70, 174–188. <https://doi.org/http://dx.doi.org/10.1016/j.ocemod.2012.12.001>
- Rueda, A., Vitousek, S., Camus, P., Tomás, A., Espejo, A., Losada, I.J., Barnard, P.L., Erikson, L.H., Ruggiero, P., Reguero, B.G., Mendez, F.J., 2017. A global classification of coastal flood hazard climates associated with large-scale oceanographic forcing. *Sci. Rep.* 7, 5038. <https://doi.org/10.1038/s41598-017-05090-w>
- Saha, S., Coauthors, 2010. The NCEP Climate Forecast System R eanalysis. *Am. Meteorol. Soc. Bull.* 91, 1015–1057. <https://doi.org/10.1175/2010BAMS3001.1>
- Vanderer, J.P., Siegel, E.M., Brubaker, J.M., Friedrichs, C.T., 2008. Influence of Spectral Width on Wave Height Parameter Estimates in Coastal Environments. *J. Waterw. Port, Coastal, Ocean Eng.* 134, 187–194. [https://doi.org/10.1061/\(ASCE\)0733-950X\(2008\)134:3\(187\)](https://doi.org/10.1061/(ASCE)0733-950X(2008)134:3(187))

Reviewer #3 (Remarks to the Author):

I have reviewed the last version of the manuscript "A recent increase in global wave power as a consequence of oceanic warming" by Reguero et al. and am now happy with the changes and updates.

I congratulate the authors for the effort, and recommend the full acceptance of the manuscript as is for publication, pending on editorial details.